# Rapid spread of a densovirus in a major crop pest following wide-scale adoption of Bt-cotton in China

Yutao Xiao[1,2†], Wenjing Li[1,3†], Xianming Yang[1†], Pengjun Xu[4,5†], Minghui Jin[2], He Yuan[2], Weigang Zheng[2], Mario Soberón[6], Alejandra Bravo[6], Kenneth Wilson[2,5], Kongming Wu[1*]

[1]The State Key Laboratory for Biology of Plant Disease and Insect Pests, Institute of Plant Protection, Chinese Academy of Agricultural Sciences, Beijing, China; [2]Shenzhen Branch, Guangdong Laboratory of Lingnan Modern Agriculture, Genome Analysis Laboratory of the Ministry of Agriculture and Rural Affairs, Agricultural Genomics Institute at Shenzhen, Chinese Academy of Agricultural Sciences, Shenzhen, China; [3]Institute of Plant Protection and Soil Fertility, Hubei Academy of Agricultural Sciences, Wuhan, China; [4]Tobacco Research Institute, Chinese Academy of Agricultural Sciences, Qingdao, China; [5]Lancaster Environment Centre, Lancaster University, Lancaster, United Kingdom; [6]Instituto de Biotecnología, Universidad Nacional Autónoma de México, Morelos, United States

**\*For correspondence:**
wukongming@caas.cn

[†]These authors contributed equally to this work

**Abstract** *Bacillus thuringiensis* (Bt) crops have been widely planted and the effects of Bt-crops on populations of the target and non-target insect pests have been well studied. However, the effects of Bt-crops exposure on microorganisms that interact with crop pests have not previously been quantified. Here, we use laboratory and field data to show that infection of *Helicoverpa armigera* with a densovirus (HaDV2) is associated with its enhanced growth and tolerance to Bt-cotton. Moreover, field monitoring showed a much higher incidence of cotton bollworm infection with HaDV2 in regions cultivated with Bt-cotton than in regions without it, with the rate of densovirus infection increasing with increasing use of Bt-cotton. RNA-seq suggested tolerance to both baculovirus and Cry1Ac were enhanced via the immune-related pathways. These findings suggest that exposure to Bt-crops has selected for beneficial interactions between the target pest and a mutualistic microorganism that enhances its performance on Bt-crops under field conditions.

## Introduction

Transgenic crops expressing insecticidal Cry proteins from *Bacillus thuringiensis* bacteria, known as Bt-crops, have become important tools for the management of insect crop pests (*Carrière et al., 2003*; *Cattaneo et al., 2006*; *Hutchison et al., 2010*; *Shelton et al., 2002*; *Tabashnik et al., 2010*). Planting of Bt-crops effectively suppresses the targeted insects, decreasing insecticide use and promoting biocontrol services (*Bravo et al., 2011*; *Carrière et al., 2003*; *Cattaneo et al., 2006*; *Hutchison et al., 2010*; *Lu et al., 2010*; *Lu et al., 2012*; *Shelton et al., 2002*; *Tabashnik et al., 2002*; *Wu, 2010*; *Wu et al., 2008*). We have previously shown that the commercialization of transgenic Bt-cotton in China brought significant changes in the ecology of insects utilizing the crop (*Lu et al., 2010*; *Swiatkiewicz et al., 2014*; *Wu et al., 2008*). However, the effect of Bt-crops on other organisms, such as microbes, which could be playing important roles in the life cycle of insect populations, remains largely unknown.

Recently, we showed in laboratory trials that infection with a densovirus, *Helicoverpa armigera* densovirus-1 (HaDV2, named as HaDNV-1 previously, GenBank accession number: NC_015718), was associated with significantly enhanced tolerance of cotton bollworm, *H. armigera*, to a baculovirus (*H. armigera* nucleopolyhedrovirus, HaNPV) (*Xu et al., 2014*), and there was some suggestion that the densovirus also increased tolerance to Cry1Ac toxin in a Bt-susceptible strain of *H. armigera* (*Xu et al., 2014*; *Xu et al., 2017a*; *Xu et al., 2017b*). HaDV2 was found to be widespread in wild populations of *H. armigera* adults (>67% prevalence between 2008 and 2012) (*Xu et al., 2014*). The densovirus was mainly distributed in the fat body of the insect and could be both horizontally and vertically transmitted. Moreover, HaDV2-positive individuals showed faster development and higher fecundity than non-infected individuals. There was no evidence for a negative effect of HaDV2 infection on *H. armigera* in relation to other fitness-related traits, suggesting a possible mutualistic interaction between the cotton bollworm and HaDV2 (*Xu et al., 2014*).

Here, we further explore the interaction between *H. armigera*, HaDV2, and Bt-cotton, to establish its relevance to field populations and to test the hypothesis that the widespread adoption of Bt-cotton in China has selected for cotton bollworm carrying the densovirus. Laboratory experiments show that HaDV2 infection in both Cry1Ac-resistant and Cry1Ac-susceptible strains of cotton bollworm enhances larval tolerance to Bt and overall fitness. Field experiments indicate that *H. armigera* populations infected with HaDV2 also have a higher tolerance to Bt-cotton relative to those not infected with HaDV2. Moreover, field monitoring over a 10-year period indicates that the frequency of HaDV2-infected cotton bollworm is significantly higher in regions planted with Bt-cotton than in those areas where Bt-cotton is not grown, and that in Bt-cotton-growing areas, the prevalence of HaDV2 infection increases with time since Bt-cotton adoption and with the proportion of cotton that is grown which is transgenic. Further, we found that increased Bt tolerance is associated with activated immune pathways to HaDV2 infection. These results indicate that increased HaDV2 infection in cotton bollworm is correlated with the wide-scale adoption of Bt-cotton in China. Our data are consistent with the notion that exposure to Bt-crops selects for beneficial interactions between the target pest and a microorganism that enhances their fitness in response to Bt exposure under field conditions.

## Results

### HaDV2 infection increases Cry1Ac tolerance in *H. armigera*

Previous laboratory bioassays suggested that when a Cry1Ac-susceptible strain of *H. armigera* was infected with HaDV2, it increased its tolerance to the Cry1Ac toxin (*Xu et al., 2014*). To explore the generality of this finding, we first analyzed the effect of HaDV2 infection in different *H. armigera* populations that differ in their susceptibility to Cry1Ac due to different mechanisms of resistance. Two susceptible *H. armigera* strains infected with HaDV2 (96S and LF) both showed 1.5 times greater tolerance to Cry1Ac toxin, relative to their corresponding non-infected controls (*Supplementary file 1a and b*). In the case of the Cry1Ac-resistant strains (BtR, 96CAD, LFC2, LF5, LF60, LF120, and LF240), infection with HaDV2 again showed a significant increase in their tolerance relative to the corresponding strains without HaDV2 infection, ranging between 30% and 130% enhanced tolerance (*Figure 1*). The slope of the regression line is significantly greater than one (t-test: t=2.853, df=6, p=0.029), suggesting that the benefits of carrying HaDV2 may increase with increasing levels of Bt tolerance. Logistic regression confirmed that for a given strain of *H. armigera*, larvae harboring HaDV2 were significantly more tolerant to Cry1Ac (GLM: Strain: $\chi^2_8$=36.57, p<0.0001, log10 (Cry1Ac toxin concentration): $\chi^2_1$=848.16, p<0.0001; Strain*log10(Cry1Ac toxin concentration): $\chi^2_8$=97.30, p<0.0001; HaDV2-status: $\chi^2_1$=9.53, p=0.0020). However, there was no evidence that the benefits of hosting HaDV2 are affected by the tolerance level of the *H. armigera* strain, as reflected in the $LC_{50}$ of the non-infected insects ($\chi^2_1$=0.06, p=0.80). At 8 days post-hatching, *H. armigera* larvae were significantly lighter (t=10.164, df=32, p<0.0001, n=17) and had lower HaDV2 viral loads in individuals feeding on diet containing Bt than the ones without Bt (t=4.527, df=32, p<0.0001, n=17), suggesting that Bt decreased the replication rate of HaDV2 by suppressing the growth of *H. armigera* larvae (*Figure 1—figure supplement 1*).

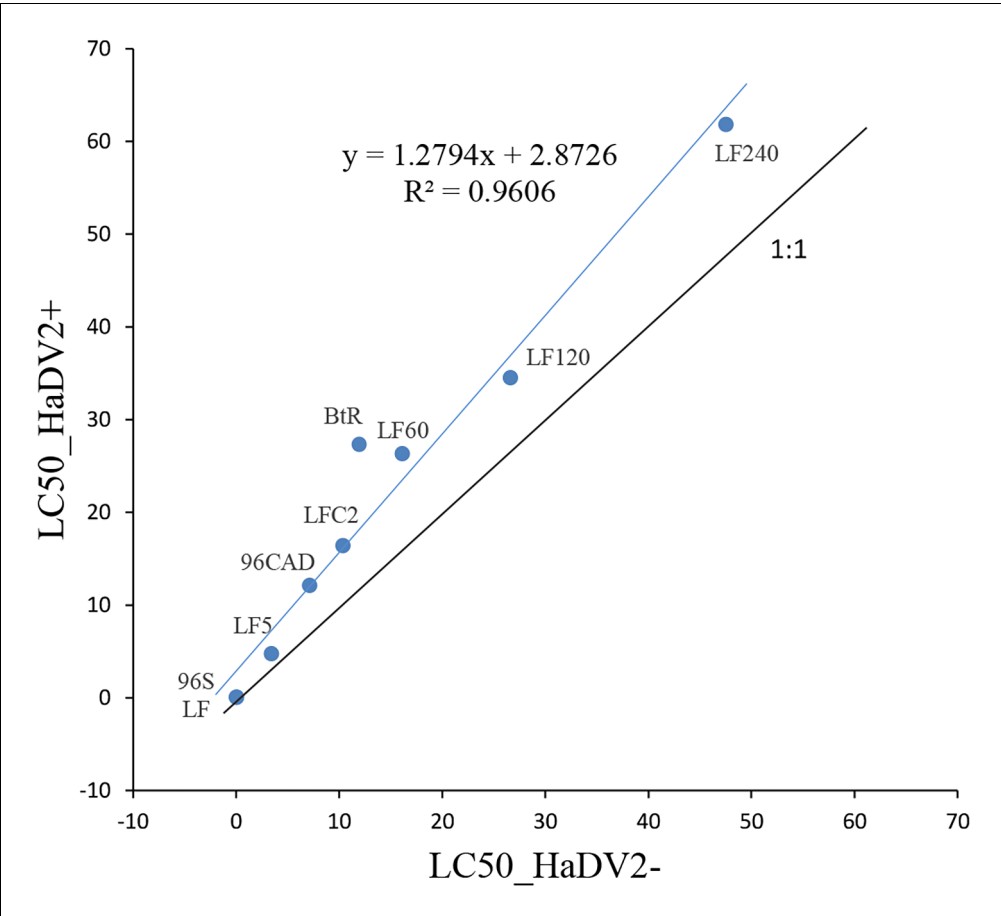

**Figure 1.** Relationship of different *Helicoverpaarmigera* strains' $LC_{50}$ with or without HaDV2 infection. The x-axis is the $LC_{50}$ of different strains (LF, 96S, LF5, LF60, LF120, LF240, LFC2, 96CAD, and BtR) without HaDV2 infection (HaDV2-negative); the y-axis is the $LC_{50}$ of different strains (LF, 96S, LF5, LF60, LF120, LF240, LFC2, 96CAD, and BtR) with HaDV2 infection (HaDV2-positive). The regression line is described by the following equation: y=1.2794x +2.8726, $R^2$=0.9606, F=11.99, df=1,7 (p<0.0085).

The online version of this article includes the following source data and figure supplement(s) for figure 1:

**Source data 1.** Source data for *Figure 1*.
**Figure supplement 1.** Quantification of HaDV2 in individuals feeding on diet with and without Bt.
**Figure supplement 1—source data 1.** Quantification of HaDV2 in individuals feeding on diet with and without Bt toxin.

## HaDV2 infection reduces the fitness cost of *H. armigera* associated with Cry1Ac-resistance evolution

To determine whether infection with HaDV2 reduces the costs associated with evolving resistance to Bt, a range of fitness traits (*Supplementary file 1c*) were measured in four strains of *H. armigera* that have different Bt-resistance levels (LF, LF5, LF60, and LF240) and were infected or not infected with HaDV2. Enhanced Bt-resistance in *H. armigera* LF, LF5, LF60, and LF240 strains was associated with lower larval survival rates, prolonged larval and pupal development (in both sexes), reduced pupal weight, lower adult emergence rate, and reduced fecundity and egg hatch rate; in contrast, sex ratio at emergence and the longevity of adults of both sexes were not influenced by the capacity to resist Cry1Ac toxin (*Figure 2*; *Supplementary file 1c and d*). When traits were combined to estimate the fundamental net reproductive rates of the four *H. armigera* strains, $R_0$ (*Wang et al., 2016*), this revealed that in the absence of HaDV2 infection, the fitness of the most Bt-resistant strain (LF240) was around 40% of that of the most Bt-susceptible strain (LF), consistent with a large fitness

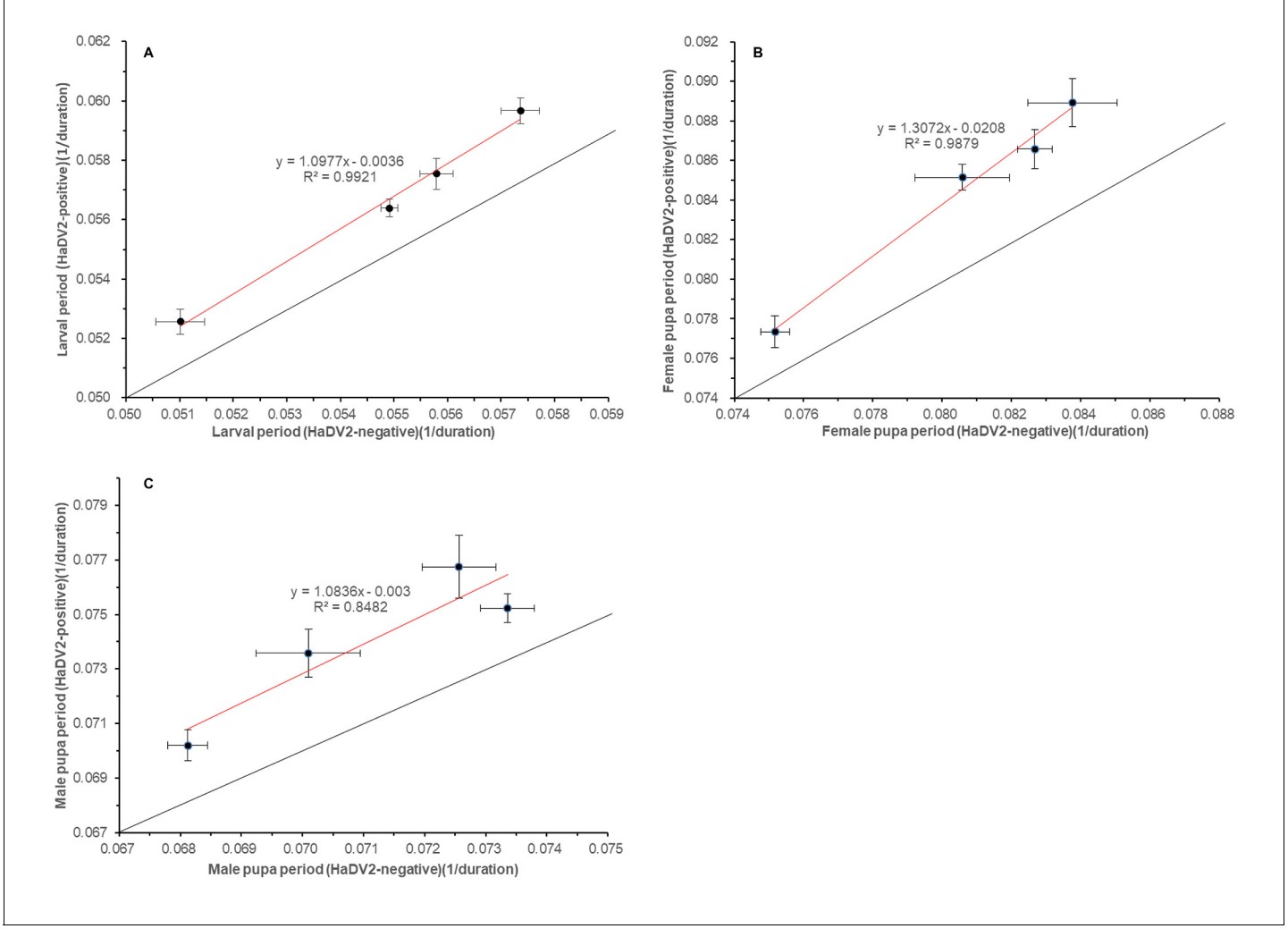

**Figure 2.** Relationship of different *Helicoverpaarmigera* strains' larval (**A**), and pupal (**B,C**) with or without HaDV2 infection. (**A**) The x-axis is the larval development rate (1/duration) of different strains (LF, LF5, LF60, and LF240) without HaDV2 infection (HaDV2-negative); the y-axis is the larval development rate (1/duration) of different strains (LF, LF5, LF60, and LF240) with HaDV2 infection (HaDV2-positive), y=1.0977x−0.0036, $R^2$=0.9921, F=176.678, df=1.2, p=0.006. (**B**) The x-axis is the female pupal development rate (1/duration) of different strains (LF, LF5, LF60, and LF240) without HaDV2 infection (HaDV2-negative); the y-axis is the female pupa period (1/duration) of different strains (LF, LF5, LF60, and LF240) with HaDV2 infection (HaDV2-positive), y=1.3072x−0.0208, $R^2$=0.9879, F=125.211, df=1.2, p=0.008. (**C**) The x-axis is the male pupa period (1/duration) of different strains (LF, LF5, LF60, and LF240) without HaDV2 infection (HaDV2-negative); the y-axis is the male pupa period (1/duration) of different strains (LF, LF5, LF60, and LF240) with HaDV2 infection (HaDV2-positive), y=1.0836x−0.003, $R^2$=0.8482, F=7.581, df=1.2, p=0.110.

The online version of this article includes the following source data for figure 2:

**Source data 1.** Source data for *Figure 2*.

cost of resistance (*Figure 3*); strains with intermediate levels of resistance (LF5 and LF60) suffered a lower cost of resistance (*c.* 30% reduction in fitness).

Infection with HaDV2 rescued, or partially rescued, this fitness loss, in all four *H. armigera* strains showing a significant increase in $R_0$ relative to their non-infected counterparts, averaging around 38% higher (paired t-test: t=4.831, df=3, p=0.017; *Figure 3*). Indeed, when infected with HaDV2, the $R_0$ values of two of the three resistant strains (LF5 and LF60) were comparable to that of the non-infected susceptible (LF) strain (*Figure 3*).

To quantify the fitness cost associated with Cry1Ac-tolerance in the more realistic context of multiple plant defenses, the growth rate of the different strains of *H. armigera* larvae was also analyzed on Bt-cotton plant leaves. Larval weight after 9 days growth was significantly affected by the cotton variety (Bt or non-Bt) and by the HaDV2 infection-status (*Supplementary file 1e and f*), with larvae

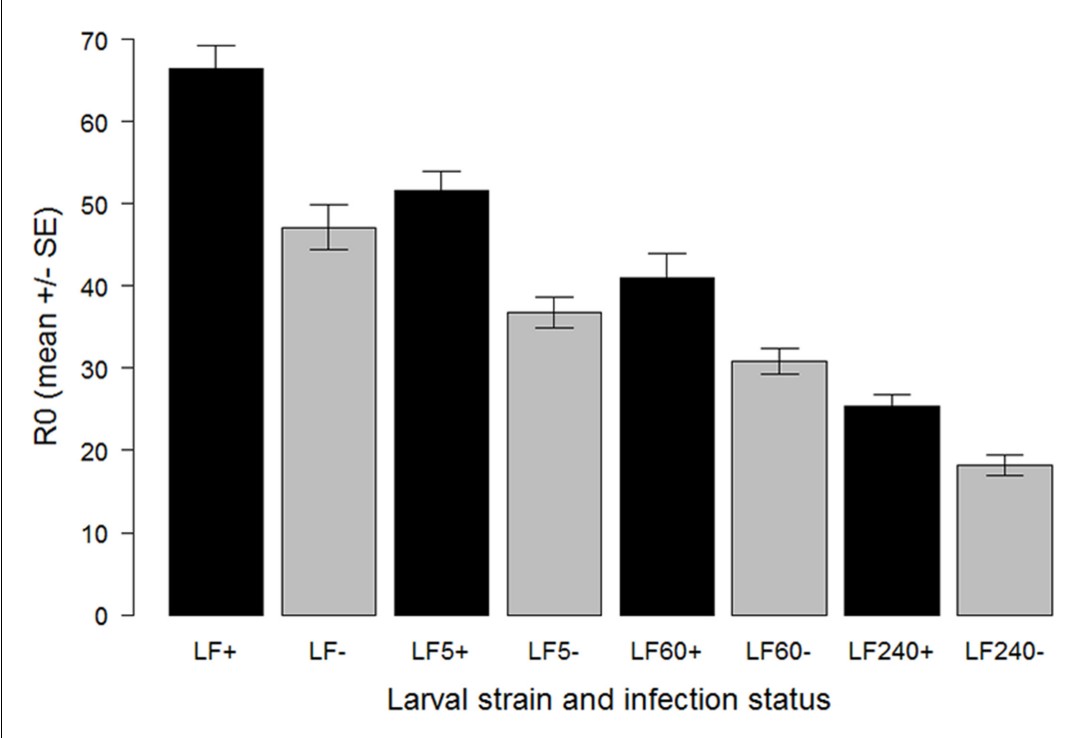

**Figure 3.** Effects of HaDV2 infection on the net reproductive rate ($R_0$) in four *Helicoverpaarmigera* strains differing in their tolerance to Bt and not exposed to Cry1Ac toxin. Mean $R_0$ is calculated as the number of female offspring per female that reaches adulthood. The bars are bootstrapped standard errors.

The online version of this article includes the following figure supplement(s) for figure 3:

**Figure supplement 1.** Predicted spread of HaDV2 on Bt-cotton for two strains of *Helicoverpaarmigera* (LF and LF240).

generally being heavier when fed with non-Bt-cotton than with Bt-cotton, and heavier for HaDV2-infected larvae than for larvae not infected with HaDV2; larvae were also heavier when they expressed lower levels of Bt-resistance, indicating that the cost of resistance is reflected in larval growth (*Supplementary file 1e and f*). None of the interactions between these three main effects explained any additional variation (model comparison with and without interaction terms: F=0.601, df=14, p=0.86), suggesting that the effects of host plant, *H. armigera* strain, and infection-status on larval growth were additive.

## HaDV2 infection levels in field populations of *H. armigera* have increased over the adoption period of Bt-cotton

To determine if infection with HaDV2 could increase the performance of *H. armigera* when exposed to Bt-cotton in the field, we collected *H. armigera* moths from Xiajin (Shandong Province) and Anci (Hebei Province) in northern China, two locations where Bt-cotton has been widely planted over the last decade (*An et al., 2015*). Across two successive years, the prevalence of HaDV2 was extremely high (98% of 637 larvae in 2015; 97% of 180 larvae in 2016). Moreover, across both years, the relative average development rates (RADRs) (*An et al., 2015*) of individuals infected with HaDV2 were significantly higher than that of larvae not infected with the virus (0.62 vs. 0.52 in 2015; 0.61 vs. 0.53 in 2016) (linear model: infection status: F=28.80; df=1.815, p<0.0001; year: F=2.10, df=1.814, p=0.15; infection*year: F=0.57, df=1.813, p=0.57) (*Figure 4*).

Given the apparent selective advantage of HaDV2 infection for insects feeding on Bt-cotton, we predicted that over time we would observe an increase in HaDV2 infection rates in the field and that this would be associated with a temporal increase in average development rates for larvae feeding on Bt-cotton plants. As predicted, over the 10-year period between 2007 and 2016, at both Xiajin and Anci provinces, HaDV2 infection rates increased significantly over time (logistic regression:

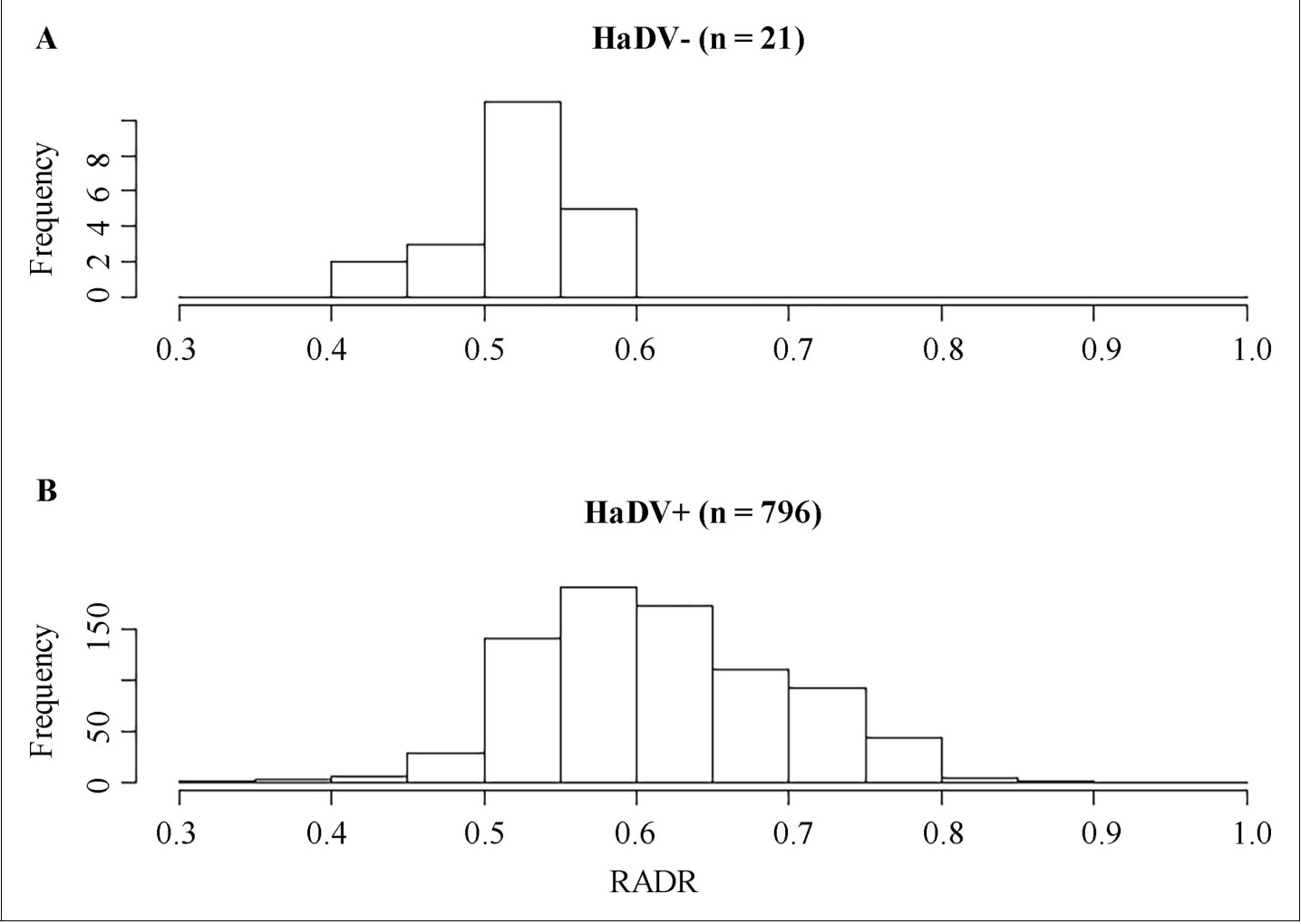

**Figure 4.** Frequency distributions for RADR scores for HaDV2 positive and negative insects. The data were collected from field-collected insects from Xiajin and Anci in 2015 and 2016. RADR, relative average development rate.

The online version of this article includes the following source data for figure 4:

**Source data 1.** Source data for *Figure 4*.

Xiajin: $\chi^2_1$=405.79, p<0.0001; Anci: $\chi^2_1$=325.21, p<0.0001) (*Figure 5A and B*). Associated with this, there was a significant temporal increase in larval development rates (RADR) at both locations (linear models: Xiajin: F=5.474, df=1.8, p=0.047; Anci: F=23.256, df=1.8, p=0.0047) (*Figure 5C and D*). Moreover, across the 10 years at both monitoring locations, there was a strong positive association between HaDV2 infection levels and RADRs, consistent with a possible causal relationship between these two temporal trends (linear models: Xiajin: F=23.826, df=1.9, P=0.001; Anci: F=13.676, df=1.9, P=0.006) (*Figure 5E and F*).

## Across regions the HaDV2 infection rates increase with increasing exposure to Bt-cotton

To further test the association between *H. armigera* densovirus infection levels and the adoption of Bt-cotton, we monitored HaDV2 infection rates at 36 locations across 16 provinces during the period 2014–2016, including locations planted with transgenic Bt-cotton (29 monitoring points across 12 provinces) and locations where Bt-cotton has not been planted (9 monitoring points across 4 provinces) (*Figure 6*; *Supplementary file 1g*). Across all 3 years, HaDV2 infection levels in *H. armigera* were significantly higher at locations where Bt-cotton was planted (mean=82%) than in those where it was not (15%) (*Figure 6—figure supplement 1*) (logistic regression: crop (Bt vs. non-Bt):

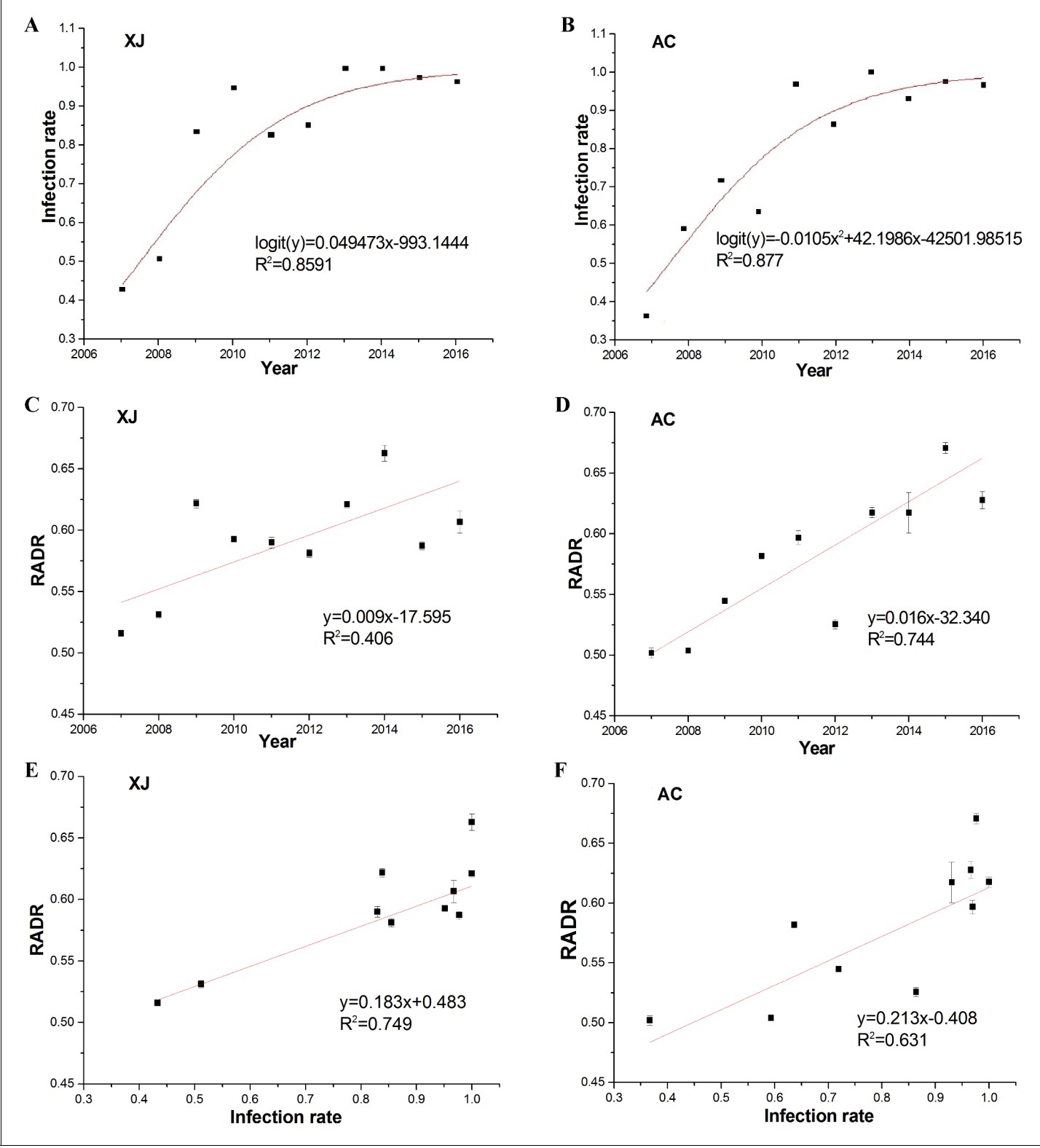

**Figure 5.** HaDV2 infection rate and RADR dynamics and their relationship for each year in the Xiajin and Anci populations during 2007–2016. (**A**) Relation between HaDV2 infection rate of larvae in Xiajin populations and planting year of Bt-cotton. Logistic regression model of HaDV2 infection rate, logit (y) = 0.49473x−993.1444, $R^2$=0.8591, $\chi^2$=405.79, df=1, p<0.0001. (**B**) Relation between HaDV2 infection rate of larvae in Anci populations and planting year of Bt-cotton. Logistic regression model of HaDV2 infection rate, logit (y) = −0.0105x²+42.1986x−42501.98515, $R^2$=0.877, $\chi^2$=325.21, df=1, p<0.0001. (**C**) Relation between RADR of larvae in Xiajin populations and planting year of Bt-cotton. Linear model of RADR, y=0.009x−17.595, $R^2$=0.406,

*Figure 5 continued on next page*

*Figure 5 continued*

F=5.474, df=1.8, p=0.047. (D) Relation between RADR of larvae in Anci populations and planting year of Bt-cotton. Linear model of RADR, y=0.016x−32.340, $R^2$=0.744, F=23.256, df=1.8, P=0.001. (E) Relationship of larvae RADR in Xiajin population and HaDV2 infection rate during the years 2007–2016, each data point is a different year, in the Linear model of RADR, y=0.183x+0.438, $R^2$=0.749, F=23.826, df=1.9, p=0.001. (F) Relationship of larvae RADR in Anci populations and HaDV2 infection rate during the years 2007–2016, each data point is a different year, in the Linear model of RADR, y=0.213+0.408, $R^2$=0.625, F=13.676, df=1.9, p=0.006. The bars are the standard error of the mean RADR for the field-derived strains tested in each year. RADR, relative average development rate.

The online version of this article includes the following source data for figure 5:

**Source data 1.** Source data for *Figure 5*.

$\chi^2_1$=354.15, p<0.0001). There was also a significant year-by-crop interaction ($\chi^2_1$=24.13, p<0.0001) due to HaDV2 infection levels being uniformly high across the 3 years at sites growing Bt-cotton (81–90%), whereas infection levels gradually increased from 2014 to 2016 at non-Bt sites (12%, 16%, and 44%, respectively).

Moreover, across the provinces where Bt-cotton is grown, the mean prevalence of HaDV2 in *H. armigera* between 2014 and 2016 increased with the number of years since Bt-cotton was first introduced ($\chi^2_1$=173.59, p<0.0001) (*Figure 7A* and *Figure 6—figure supplement 1*) and also increased as the proportion of cotton that was transgenic increased ($\chi^2_1$=5.34, p=0.021) (*Figure 7B*). HaDV2 prevalence was not correlated with the proportional area of any of the other crops grown ($\chi^2_1$<1.310, p>0.25). Adding environmental variables to this minimal model, either singly or in combination, did not significantly improve the model fit (average rainfall: $\chi^2_1$=0.110, p=0.74; average temperature: $\chi^2_1$=0.155, p=0.69; average altitude: $\chi^2_1$=0.001, p=0.98; rainfall+temperature+altitude: $\chi^2_3$=0.572, p=0.90). These results are consistent with the notion that the benefits of HaDV2 infection are greatest for *H. armigera* exposed to Cry1Ac-producing cotton and that selection favoring HaDV2 infection increases the longer the insects are exposed to the Bt-cotton.

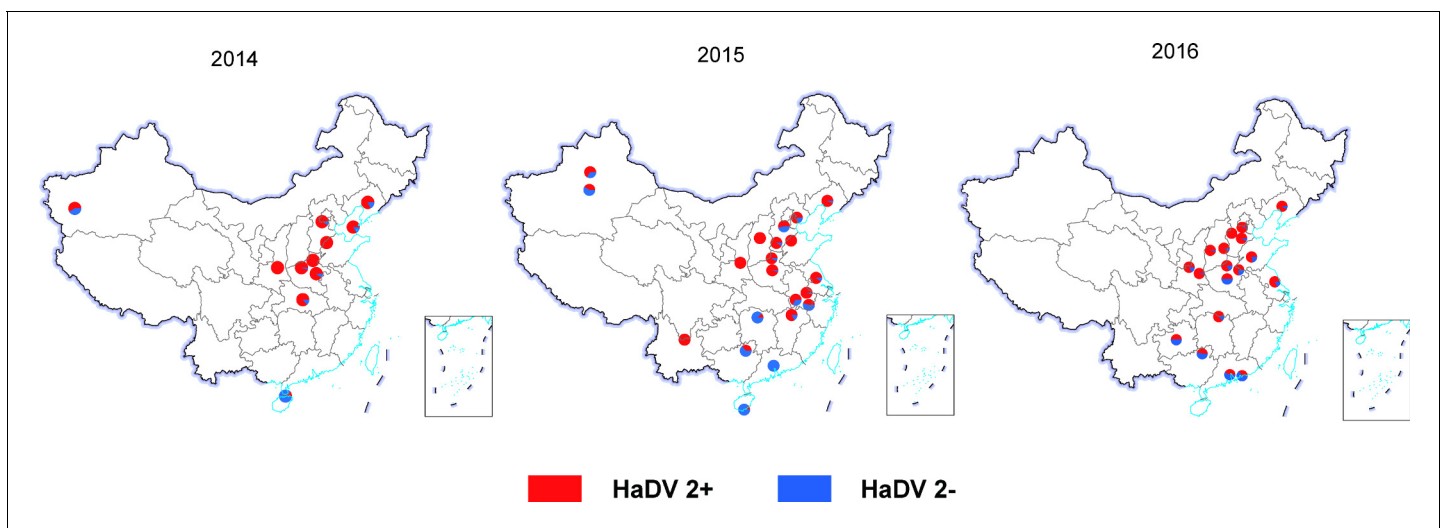

**Figure 6.** Distribution of HaDV2 in *Helicoverpaarmigera* from different populations. The red proportion of circles refers to infected individuals, and the blue refers to non-infected individuals. There are significant differences in HaDV2 infection rates between the 29 Bt-cotton planting points and 7 non-Bt-cotton planting points (code: 12, 29, 30, 31, 32, 49, and 50). The sample information was summarized in *Supplementary file 1g*.

The online version of this article includes the following figure supplement(s) for figure 6:

**Figure supplement 1.** The infection rate of HaDV2 in Bt-cotton and non-Bt-cotton planting areas from 2014 to 2016 (means± SE).

**Figure supplement 2.** The detection limit of HaDV2 with PCR method.

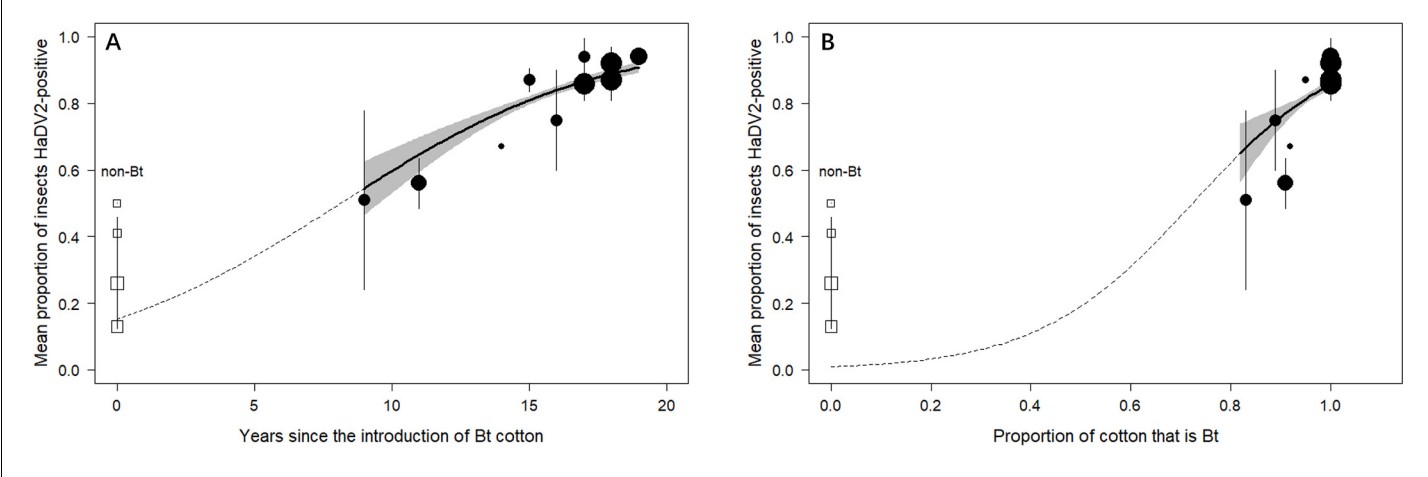

**Figure 7.** HaDV2 infection rate has increased significantly since Bt-cotton was first introduced and was positively related to the Bt adoption in all cotton areas. (**A**) Temporal changes in the infection rate of HaDV2 since the introduction of Bt-cotton. (**B**) Changes in the infection rate of HaDV2 according to the proportion of Bt-cotton in all cotton. Each symbol represents an individual province sampled for densovirus over three years (2014–2016); the mean virus prevalence (± standard error) over those 3 years is shown. Symbol size reflects sampling effort and represents data from >1500 insects. Circles represent the 12 provinces where Bt-cotton is grown; squares are the 4 provinces where Bt-cotton is not grown. The solid line represents the logistic regression (± standard error, shaded zone) describing the relationship between virus prevalence and years since the introduction of Bt-cotton to a province for the 12 Bt-cotton-growing provinces only. The dashed line extrapolates this regression line to Year 0. The detailed information is summarized in *supplementary file 1g* and *Figure 6—figure supplement 1*.

## HaDV2 infection activated the immune pathways in the cotton bollworm

To try to understand better the mechanisms increasing the Cry1Ac tolerance levels and enhanced fitness of HaDV2-infected insects, we conducted an RNA sequencing experiment (*Supplementary file 1h*).

The principal component analysis of the transcriptome with differentially expressed genes (DEGs) data clearly distinguished HaDV2-positive from HaDV2-negative individuals at three different time points: 24, 48, and less so at 72 hr after HaDV2 inoculation (*Figure 8A,B,C*). Taken together with the hierarchical clustering of these DEGs, these results suggest that the HaDV2 has a major effect on the gene expression profiles of their hosts.

We performed pathway enrichment analysis on the DEGs, focusing particular attention on pathways related to the development and immune systems (*Figure 8D,E,F*). Genes in Jak-STAT immune signaling pathway, which are related to some antiviral and antibacterial mechanisms, were significantly enriched and upregulated in the HaDV2-infected larvae at 24 and 48 hr (*Figure 8D,E* and *Figure 8—figure supplement 1A,B*), but not at 72 hr (*Figure 8F*). Interestingly, genes in ABC transporters pathway at 48 hr (*Figure 8—figure supplement 1C*), the mitogen-activated protein kinase (MAPK) signaling and lysosome pathways at 72 hr (*Figure 8—figure supplement 1D,E*), which are related to antimicrobial immune response, are significantly enriched and upregulated (*Figure 8E,F*). Genes in pathways related to development were also significantly enriched, for example, insulin, the mammalian target of rapamycin (mTOR), AMP-activated protein kinase (AMPK) signaling, and the insect hormone biosynthesis pathways at 24 hr (*Figure 8—figure supplement 1F,G,H,I*), steroid hormone biosynthesis and insulin signaling pathways at 48 hr (*Figure 8—figure supplement 1J,K*), and the mTOR, protein digestion and absorption, and steroid hormone biosynthesis pathways at 72 hr. These results may help to explain why HaDV2-positive individuals developed more quickly than non-infected insects.

## HaDV2 decreased the effect of Bt on *H. armigera*

There was a total of 1573 significant DEGs in *HaDV2-negative* insects after exposure to Cry1Ac (673 upregulated and 900 downregulated). We focused on DEGs and pathways related to Bt tolerance and immune systems. Seven ABC transporter genes, which are related to immunity, were

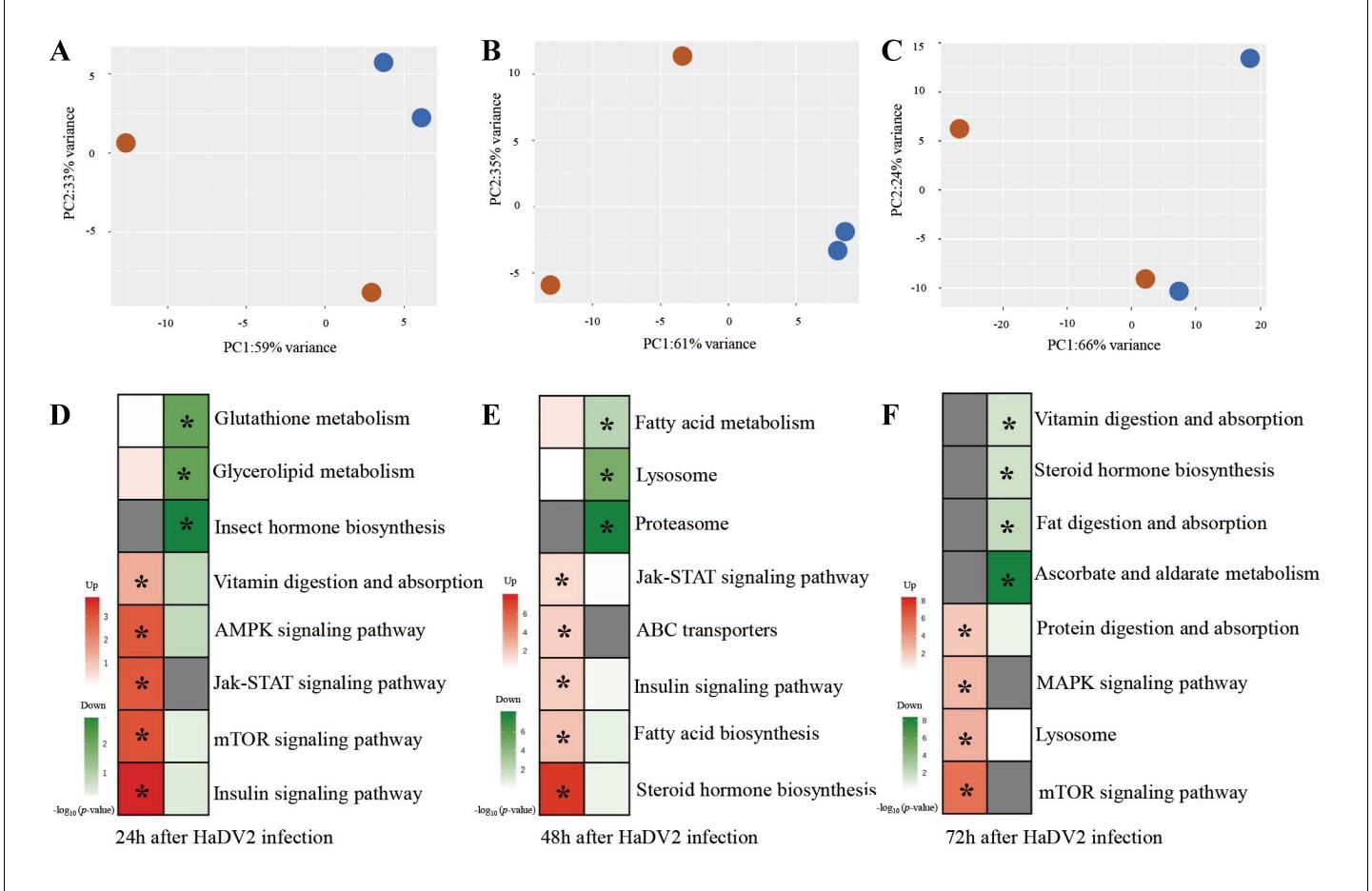

**Figure 8.** Transcriptome analysis of HaDV2-positive individuals compared to related HaDV2-negative individuals (HaDV2+ vs.HaDV2−) of *Helicoverpaarmigera*. (**A–C**) PCA of global gene expression of DEGs at 24 (**A**), 48 (**B**), and 72 hr (**C**) after HaDV2 inoculation. Blue stands for HaDV2-positive samples and red stands for HaDV2-negative samples. (**D–F**) Heatmaps of –log$_{10}$ p-values of KEGG pathways representing the upregulated and downregulated DEGs at 24 (**D**), 48 (**E**), and 72 hr (**F**). '*' indicates the significantly enriched pathways (p<0.05). Red color shows upregulation pathways, green color shows downregulation pathways, gray color shows no value, the redder/greener the color, the lower p-values. DEG, differentially expressed gene; PCA, principal component analysis.

The online version of this article includes the following figure supplement(s) for figure 8:

**Figure supplement 1.** Heatmaps of DEGs related to the expression of significantly enriched pathways of *Helicoverpa armigera* at 24, 48, and 72 hr.

differentially expressed (four upregulated and three downregulated) (*Figure 9A*); eight trypsin genes, which are related to the conversion of the protoxin to activated toxin, were differently expressed (six upregulated and two downregulated) (*Figure 9B*); two carboxylesterase genes, which are related to the detoxification of Bt by the insect, were upregulated; and two Bt toxin receptors genes, alkaline phosphatase (ALP) and aminopeptidase N (APN), which are related to Bt resistance, were downregulated (*Figure 9C*). Genes in the MAPK signaling pathway, which is related to antimicrobial immune response, was also significantly downregulated (*Figure 9D*).

In contrast, there were only 249 significant DEGs in *HaDV2-positive* insects after exposure to Cry1Ac (165 upregulated and 84 downregulated). One trypsin gene was downregulated; genes in the ascorbate and aldarate metabolism pathway, which is related to carbohydrate metabolism were upregulated; genes in drug metabolism – cytochrome P450 pathway, metabolism of xenobiotics by cytochrome P450 pathway and drug metabolism – and other enzymes pathways, which are related to detoxification, were also upregulated (*Figure 9E*).

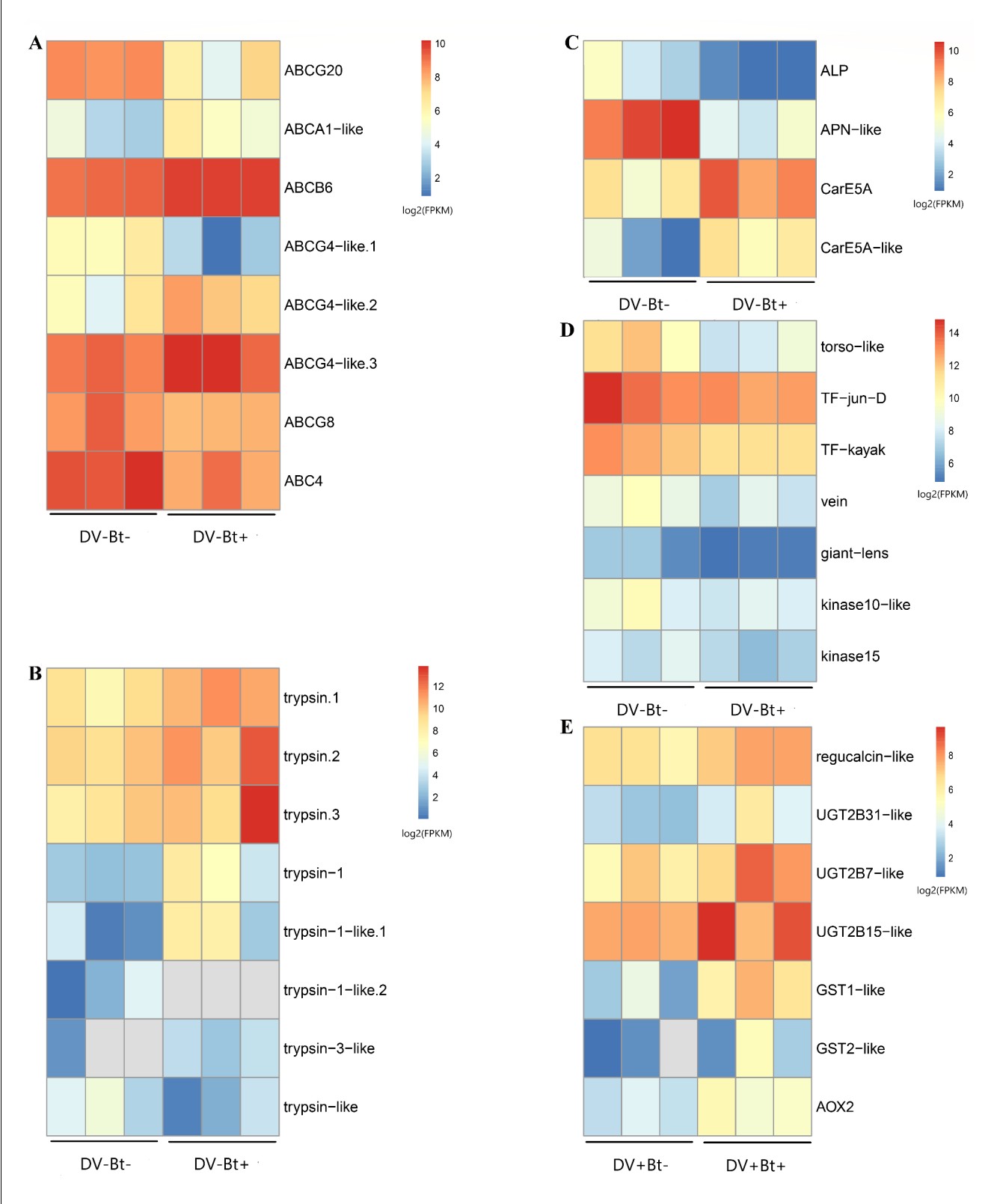

**Figure 9.** Transcriptome analysis of *Helicoverpa armigera* after HaDV2 infection and Cry1Ac exposure. The quantity of DEGs with log$_2$(FPKM) related to the expression of (**A**) the ABC transporters; (**B**) trypsin; (**C**) Bt receptors and carboxylesterase genes; (**D**) the MAPK signaling pathway; (**E**) the drug metabolism pathways. DV−=HaDV2-negative individuals, DV+=HaDV2-positive individuals. Bt−=larvae were fed on the artificial diet without Cry1Ac, Bt +=larvae were fed on the artificial diet containing 1 μg/mL Cry1Ac. Colors in log$_2$(FPKM) indicate the gene expression levels, the hotter (redder) the

*Figure 9 continued on next page*

*Figure 9 continued*

color, the higher the gene expression level. The three columns represent three biological replicates. DEG, differentially expressed gene; MAPK, mitogen-activated protein kinase.

## Discussion

Commercialization of transgenic Bt-crops has brought some significant benefits to farmers (*Carrière et al., 2003*; *Cattaneo et al., 2006*; *Lu et al., 2012*; *Shelton et al., 2002*; *Wu et al., 2008*). These Bt-plants successfully control target insect pests and have resulted in the reduced use of insecticides in the field (*Bravo et al., 2011*; *Lu et al., 2012*; *Wu et al., 2008*). Here, we show that HaDV2 infection enhances tolerance to Cry1Ac in both susceptible and resistant strains of *H. armigera*. We also show that there appears to have been a strong positive selection for HaDV2-infected *H. armigera* in populations exposed to Cry1Ac-cotton in the field, especially (but not exclusively) when this exposure has been for prolonged periods of time. Intensive monitoring in two provinces where Bt-cotton has been grown over a number of years showed a strong temporal increase in HaDV2 infection levels, from around 40% in 2007 to nearly 100% in 2016, with high levels of HaDV2 infection being associated with higher larval growth rates on Bt-cotton in the field. Indeed, at locations where Bt-cotton is not grown, *H. armigera* shows much lower rates of HaDV2 infection (12–44% vs. 81–90%), consistent with weaker selection for HaDV2-infected *H. armigera* in the absence of Bt-crops. Moreover, the prevalence of HaDV2 infection increased with the number of years since Bt-cotton was first introduced to a province and with the proportion of cotton grown that is transgenic, consistent with Bt-cotton being a key selection pressure. There were no significant relationships between HaDV2 prevalence and the proportional cover of any other main crops grown locally (cotton, rice, corn, wheat, beans, tubers, oil-producing crops, and vegetables) or with any of the environmental variables tested (rainfall, temperature, and altitude). Thus, our data suggest that the Cry1Ac-toxin selected for HaDV2-infected *H. armigera* following exposure to Bt-cotton and our RNA-Seq analysis suggests that the increased tolerance of HaDV2-infected insects is caused, in part at least, by the activation of a series of immune pathways and pathways that enhance development.

The emergence and rapid spread of a symbiont through a host population have been observed previously (*Himler et al., 2011*; *Turelli and Hoffmann, 1991*; *Weeks et al., 2007*). For example, the bacterial symbiont, *Rickettsia* sp. nr. *bellii*, swept through a population of invasive sweet potato whitefly, *Bemisia tabaci*, in just 6 years (*Himler et al., 2011*). Although we do not know when HaDV2 first infected *H. armigera* in China, in two areas where Bt-cotton is grown extensively, and where nearly 100% of all insects are currently infected with HaDV2, the prevalence of the densovirus was as low as 40% in 2007, when the earliest samples were collected (*Figure 5*). Moreover, extrapolation of the observed temporal trends (*Figure 5A,B* and *Figure 7*) would suggest that emergence of HaDV2 may have coincided with the introduction of Bt-cotton in China two decades ago. This apparently rapid spread is perhaps not surprising given the strong fitness benefits of carrying the densovirus (amounting to a ~40% increase in *H. armigera* $R_0$) (*Figure 3*). Consistent with this, based on these estimated increases in $R_0$, a simple population growth model (*Himler et al., 2011*) suggests that HaDV2 would go to near fixation in 20–30 generations in Bt-cotton-growing areas (*Figure 3—figure supplement 1*).

Rather than being a recent introduction to *H. armigera* in China, it is possible that the association between the HaDV2 and its host is more ancient, but that the increase in HaDV2 prevalence in recent years is because it has only recently evolved from a parasitic or commensal relationship into one that is more mutualistic (*Weeks et al., 2007*), perhaps under direct or indirect selection from Bt-cotton. Indeed, most of the densoviruses that have been studied thus far have tended to be at the parasitic end of the symbiotic spectrum and some of these have even been promoted as potential biological pesticides (*El-Far et al., 2012*). However, the observation that densoviruses tend to be parasitic in nature is likely biased by the fact that viruses with notable pathological effects on their hosts are more likely to be reported (*Roossinck, 2011*; *Roossinck, 2015*; *Webster et al., 2015*; *Xu et al., 2020*). With the advent of modern molecular methods, such as suppression subtractive hybridization and RNA-seq, it has become apparent that there are many 'good viruses' that have failed to be discovered due to their relatively benign, or even beneficial, effects on their hosts (*Roossinck, 2011*; *Roossinck, 2015*; *Webster et al., 2015*; *Xu et al., 2020*), including HaDV2 (*Xu et al.,*

*2014*). Although there is an increasing number of studies suggesting that symbiotic microbes can impact an insect host's ability to combat pathogens and parasites (*Graham et al., 2012*; *Hedges et al., 2008*; *Oliver et al., 2003*; *Xu et al., 2020*), as far as we are aware, this is the first field study to provide support for the notion that a symbiotic virus could protect a crop pest from Bt toxin.

Regardless of whether or not larvae are exposed to Cry1Ac, HaDV2 appears to enhance the fitness of both Bt-resistant and Bt-susceptible *H. armigera* (*Xu et al., 2012*; *Figure 3*). Theoretically, the infection rate should naturally increase in the *H. armigera* population both in Bt-cotton and non-Bt-cotton regions because of efficient transmission with horizontal and vertical modes and the mutualistic relationship between HaDV2 and its host, especially increasing its reproduction (*Xu et al., 2014*). We monitored HaDV2 infection rates at 38 locations across 16 provinces during the period 2014–2016. Indeed, our data showed that HaDV2 infection levels increased both in Bt-cotton and non-Bt-cotton areas. However, with the exception of samples from Changde (Hunan Province), the other 37 locations showed consistent trends that HaDV2 infection rates were significantly higher in Bt-cotton areas than in non-Bt-cotton areas. Previously, we also quantified the infection rate of HaDV2 in samples collected from Changde in 3 years (*Xu et al., 2012*). The final infection rate in samples from Changde in the 4 years was 45%, which is higher than the infection rate of samples from non-Bt-cotton planting regions (26%). Changde is located at the southernmost point of Bt-cotton-growing range, where the planted cotton included both non-Bt- and Bt-cotton and the fluctuation of HaDV2 infection rate in Changde might be due to both the lower pressure of Bt-cotton and the migration of *H. armigera* between Bt- and non-Bt-cotton-growing areas. Previously, we showed that there was evidence for a significant decline in HaDV2 prevalence from 2008 to 2012, especially in the migrating moths collected in Yantai (Beihuang Island), which is located in Bohai Sea between Bt- and non-Bt-cotton areas and where there were no resident populations of *H. armigera* (*Feng et al., 2004*; *Feng et al., 2005*; *Feng et al., 2009*; *Xu et al., 2014*). If we exclude the data from Yantai, however, there was no obvious temporal trend in HaDV2 prevalence between 2008 and 2012 (85%, 77%, 83%, 73%, and 75%, respectively). Other factors might also result in variation in HaDV2 infection rates. For example, there are four generations of *H. armigera* per year in northern China and the fourth generation overwinters as diapausing pupae, and if *H. armigera* grows too fast due to HaDV2 infection, it will overwinter at an inappropriate state which may lead to higher mortality of HaDV2-positive individuals (*Mu et al., 1995*; *Zhang and Li, 2001*).

The climate and crop structures of the locations where our samples were collected have been relatively constant over the last few decades and data analyses supported the notion that HaDV2 prevalence was not correlated with environmental variables or the areas of any other crops being grown. Using the PCR method, it is not possible to discount the possibility that HaDV2 was widespread in field populations of *H. armigera* as very low-titer covert infections, and that the higher incidence of virus in areas of Bt-cotton was due to the stress associated with larvae feeding on plants expressing Bt toxin. To test the hypothesis, using larvae with low dose of HaDV2 from single pair, we quantified the copy numbers of HaDV2 within the cotton bollworms feeding on artificial diets with or without Bt toxin. Interestingly, the results showed that feeding on the Bt toxin resulted in lower titers of HaDV2, which might be due to the Bt suppressing the growth of *H. armigera* larvae. These results suggest that the higher infection rate of HaDV2 might be due to lower mortality by higher Bt tolerance and the increased reproduction of the cotton bollworms with HaDV2 than those without. These findings support the notion that Bt-cotton has played an important role in increasing HaDV2 infection rates in field populations of *H. armigera*, and suggest the possibility that HaDV2 may threaten the future control of *H. armigera* by Bt.

We found some evidence to support this hypothesis from $LC_{50}$-bioassays using synthetic diets containing Cry1Ac protoxin, with the benefits of hosting HaDV2 being greatest for the most Bt-resistant strains (*Figure 1*), suggesting synergistic effects of Bt-resistance and HaDV2 infection. However, laboratory trials using non-Bt- and Bt-cotton plants suggested that the relative fitness benefits of harboring HaDV2 were independent of exposure to Bt-cotton and of Bt-resistance levels (at least in terms of larval growth rates) (*Supplementary file 1f*). This is at odds with the observation that HaDV2 infection levels in the field are higher when populations are exposed to Bt-cotton for prolonged periods, and may indicate that LC50-bioassays are a more reliable measure of relative fitness than larval growth on Bt-cotton plants in the laboratory. Alternatively, it is possible that there are added benefits to HaDV2 infection for *H. armigera* exposed to Bt-cotton in the field that our

laboratory bioassays cannot fully capture, such as the enhanced capacity to resist predators and parasitoids, or reduced exposure to chemical pesticides. In this latter regard, there may be parallels here with the increase in outbreaks of mirid bugs (Heteroptera: Miridae) following the wide-scale adoption of Bt-cotton in China and the subsequent reduction in pesticide application rate on crops (*Lu et al., 2010*; *Zhang et al., 2018*).

The evolution of tolerance to Bt-toxins could endanger the utility of this GM technology (*Gahan et al., 2001*, *Gahan et al., 2007*; *Gahan et al., 2010*; *Heckel, 2012*; *Tabashnik, 2015*; *Tabashnik et al., 2008*; *Tabashnik et al., 2013*). However, as observed here, resistance to Cry toxins is often associated with significant fitness costs that are likely to delay the establishment of Bt-resistant populations under field conditions (*Gassmann et al., 2009*; *Horikoshi et al., 2016*; *Liu et al., 2017*; *Raymond et al., 2005*; *Santos-Amaya et al., 2017*). Interestingly, we show here that Bt-resistant strains with different resistance mechanisms, such as mutations affecting ABCC2 (strain LF60) or cadherin (strain 96CAD) (*Liang et al., 2008*; *Xiao et al., 2014*), incurred significantly reduced fitness costs when they were infected with HaDV2. The reduction of fitness costs by HaDV2 infection, especially increasing the reproduction of their hosts, could speed up the establishment of resistant populations. The Bt resistance induced by ABCC2 and cadherin is due to genomic heritable variation (*Liang et al., 2008*; *Xiao et al., 2014*), while the Bt tolerance induced by HaDV2 appears to be due to the regulation of immune and other pathways. ABCC2 and cadherin can directly mediate high levels of resistance to Bt toxin in *H. armigera*, but with significant fitness costs (e.g., lower reproduction) (*Cao et al., 2014*); however, HaDV2 infection could enhance host reproduction and tolerance to Cry1Ac in both susceptible and resistant strains of *H. armigera*, increasing larval survival rate. Susceptible strains infected with HaDV2 showed only 1.5 times greater tolerance to Cry1Ac toxin. In the case of the Cry1Ac-resistant strains, infection with HaDV2 again showed a significant increase in their resistance relative to the corresponding strains without HaDV2 infection, ranging between 30% and 130% enhanced resistance. Thus, a potential implication of these results is that the increased prevalence of HaDV2 infections under field conditions could impact the evolution of *H. armigera* populations resistant to Bt-cotton, which might decrease the control efficacy by Bt-cotton expressing Cry1Ac toxin and promote the development of novel strategies for *H. armigera* control.

The temporal increase in HaDV2 infection levels observed in Xiajin and Anci, two cotton-growing counties >300 km apart in north-east China, was associated with parallel temporal increases in relative growth rates (RADR) (*Figure 5C,D*). In a previous study of these two populations, the temporal increase in RADR between 2002 and 2014 was ascribed to evolved genetic resistance to Cry1Ac in *H. armigera*, despite the failure to identify the key genetic locus conferring resistance in these populations (*An et al., 2015*). The relatively strong correlation between the annual prevalence of HaDV2 in *H. armigera* and their mean annual RADR in each of these populations (*Figure 5E,F*) suggests that the putative increase in genetic resistance over this period may, at least in part, be due to the phenotypic expression of HaDV2 infection. The fact that the densovirus is mostly located in the insect's fat body suggests that it may play a role in *H. armigera* development (*Xu et al., 2014*). Lepidopteran larvae tend to become more tolerant to pathogens and toxins as they age and grow, a phenomenon known as developmental resistance (*Engelhard and Volkman, 1995*). HaDV2-infected *H. armigera* not only accumulate more fat body than non-infected individuals, but they also grow faster and larger (*Xu et al., 2012*). Therefore, developmental resistance could provide a potential mechanism to explain the enhanced tolerance of densovirus-infected insects. HaDV2-infected larvae have more fat body than non-infected insects (*Xu et al., 2014*), and the fat body plays an important role in insect immunity as it is a major site for the production and secretion of antimicrobial peptides (*Hoffmann, 2003*; *Lemaitre and Hoffmann, 2007*), so immunological resistance is another potential mechanism by which HaDV2 may enhance resistance to both baculovirus and Cry1Ac.

To explore further the molecular interaction between the HaDV2 and its host, we performed an RNA-seq experiment using *H. armigera* larvae. Due to the observed effects of HaDV2 on its host, we focused on molecular pathways related to immunity and development (*Jindra et al., 2013*; *Kingsolver and Hardy, 2012*; *Kingsolver et al., 2013*; *Lin and Smagghe, 2019*; *Shields, 2017*). In *H. armigera* larvae at 24 and 48 hr, the Jak-STAT pathway was significantly enriched by upregulation in HaDV2-positive individuals. In fruit flies and mosquitoes, this pathway controls the expression of genes in response to infection with a range of viruses and bacteria, including *Drosophila* C virus, dengue virus, West Nile virus, *Escherichia coli*, and *Micrococcus luteus* (*Barillas-Mury et al., 1999*;

*Marques and Imler, 2016*). The Jak-STAT pathway was not significantly enriched in larvae at 72 hr, however, pathways related to other antimicrobials were, for example, the lysosome and MAPK signaling pathways (*Takano et al., 2019*; *Watts, 2012*). Previously, we found that HaDV2-positive individuals developed faster than non-infected insects (*Xu et al., 2014*). Interestingly, the transcriptome data suggest that several pathways related to development are also significantly enriched, including the insect hormone synthesis, insulin, mTOR, and AMPK signaling pathways (*Bland et al., 2010*; *Jindra et al., 2013*; *Lin and Smagghe, 2019*; *Saxton and Sabatini, 2017*). We did not observe the differential expression of genes playing pivotal roles in Bt resistance of *H. armigera*, for example, cadherin, APN, and ALP (*Chen et al., 2015*; *Gahan et al., 2001*; *Zhang et al., 2009*), suggesting that HaDV2 may not enhance resistance to Bt using traditional molecular resistance pathways, though we did observe that the ABC transporter pathways, which has been reported to be associated with Bt-resistance (*Bretschneider et al., 2016*; *Dermauw and Van Leeuwen, 2014*), were significantly upregulated at 48 hr post challenge.

Generally, the mode of action of Bt toxins agrees that Bt protoxins are first converted to activated toxins by midgut proteases, then the toxins bind to the midgut receptors, for example, CAD, APN, and ALP, finally leading to insect death (*Xiao and Wu, 2019*). A change in any of these steps will lead to Bt resistance, such as mutations in genes for toxin activation, toxin-binding, and changes in insect immunity (*Xiao et al., 2014*; *Zhang et al., 2009*). To understand the responses of *H. armigera* to Bt toxins with or without HaDV2, we conducted RNA-seq after larvae had been exposed to Cry1Ac. In insects lacking HaDV2, Cry1Ac increased the activation of Bt protoxins by increasing the expression level of proteases, such as trypsin. In a defensive response to Cry1Ac, receptor genes of *H. armigera*, such as APN and ALP, were significantly downregulated to reduce the toxin-binding capacity. Carboxylesterase genes and most of ABC transporters genes were significantly upregulated to improve insect immunity, while other pathways related to antimicrobials, for example, MAPK signaling pathways, were significantly downregulated, reducing resistance to Bt toxin (*Yang et al., 2020*). In contrast, when *H. armigera* larvae were infected with HaDV2, these genes and pathways were generally not significantly enriched after Cry1Ac exposure: one trypsin gene was downregulated and some genes involved in drug metabolism were upregulated to increase the metabolism of xenobiotics, improving Bt tolerance. Through RNA-seq analysis, we did not find significant upregulation or downregulation of genes known to be related to typical Bt tolerance. Thus, it is difficult to perform further experiments, e.g. CRISPR. In spite of this, our data still support the notion that HaDV2 increased the Bt tolerance level of *H. armigera* via related immune mechanisms.

Whilst data from the field are inevitably correlational, when combined with the laboratory experiments, we feel that the most parsimonious explanation for the observations reported here is that HaDV2 directly or indirectly enhances tolerance of *H. armigera* to Bt in the field, probably via increasing larval growth rates and via priming of immune defenses. However, demonstrating causation conclusively in the field will prove difficult. One possibility would be to plant some Bt-cotton in areas that are currently Bt-free and to compare larval growth patterns and HaDV2 infection levels in neighboring Bt and non-Bt areas, mirroring the 'natural' field experiments we report here. As was found with the rapid spread of mutualistic *Rickettsia* in whiteflies (1–97% prevalence in just 6 years or around 80 field generations; *Himler et al., 2011*), we envisage that HaDV2 infection levels will eventually go to fixation (or close to it) in both Bt and non-Bt areas, due to the growth benefits associated harboring this mutualist, but that the time to get to this point will take longer in non-Bt areas, due to the lower fitness benefits. Likewise, we envisage that the spread of HaDV2 infection would slow down in Bt areas if Bt-cotton were to be replaced by conventional cotton or a GM variety that did not rely on Bt toxins. Another possibility would be to conduct laboratory evolution experiments using populations of *H. armigera* infected or not with HaDV2 growing on either Bt- or non-Bt-cotton over several generations. We envisage that HaDV2 infection levels would increase over time on both Bt- and non-Bt-cotton due to the fitness benefits of HaDV2 infection, but that the rate of spread would be faster on Bt-cotton. We also envisage that the fitness of *H. armigera* ($R_0$ and cumulative population size after several generations) would be determined by the interaction between HaDV2 infected-status and cotton plant Bt-status. A similar experiment to this was conducted by *Himler et al., 2011* for laboratory cages of whitefly populations infected with *Rickettsia*, who found that over five generations, the prevalence of *Rickettsia* increased from 14% in the parental generation to 40–70% on three different host plant species. Repeating such experiments with *H. armigera* is feasible but logistically much more difficult due to their much larger size. In summary, here, we

report the rapid increase in the prevalence of a mutualistic virus in a major crop pest and show that its spread is greatest for field populations exposed to Bt-cotton for prolonged periods. Our findings support a novel mechanism by which insects may cope with changes in engineered host plant defenses, namely via infection with a mutualistic virus that enhances their tolerance to the toxins expressed by Bt-cotton crops. Our study has significant implications for understanding the co-evolution of host-pathogen-symbiont interactions. Although we show that HaDV2 increases the tolerance of its host to Bt, it is not feasible to manipulate HaDV2 infection status in the field in order to enhance the capacity of Bt to manage *H. armigera* because of the lack of suitable antivirals (against HaDV2) and because of the efficient vertical and horizontal transmission of this virius in the field.

# Materials and methods

## Key resources table

| Reagent type (species) or resource | Designation | Source or reference | Identifiers | Additional information |
|---|---|---|---|---|
| Strain, strain background (*Helicoverpa armigera*) | LF | Collected from Langfang, Hebei Province, in 1998 | | See Materials and methods, Laboratory strains |
| Strain, strain background (*H. armigera*) | 96S | Collected from Xinxiang, Henan Province, in 1996, | | See Materials and methods, Laboratory strains |
| Strain, strain background (*H. armigera*) | BtR | Cry1Ac-resistant strains selected from the susceptible strain on artificial diets | | See Materials and methods, Laboratory strains |
| Strain, strain background (*H. armigera*) | 96CAD | Cry1Ac-resistant strains (with a cadherin mutation) selected from the susceptible strain on artificial diets | | See Materials and methods, Laboratory strains |
| Strain, strain background (*H. armigera*) | LFC2 | Cry1Ac-resistant strains (with an ABCC2 mutation) selected from the susceptible strain on artificial diets | | See Materials and methods, Laboratory strains |
| Strain, strain background (*H. armigera*) | LF5 | Cry1Ac-resistant strains selected from the susceptible strain on artificial diets | | See Materials and methods, Laboratory strains |
| Strain, strain background (*H. armigera*) | LF30 | Cry1Ac-resistant strains selected from the susceptible strain on artificial diets | | See Materials and methods, Laboratory strains |
| Strain, strain background (*H. armigera*) | LF60 | Cry1Ac-resistant strains selected from the susceptible strain on artificial diets | | See Materials and methods, Laboratory strains |
| Strain, strain background (*H. armigera*) | LF120 | Cry1Ac-resistant strains selected from the susceptible strain on artificial diets | | See Materials and methods, Laboratory strains |
| Strain, strain background (*H. armigera*) | LF240 | Cry1Ac-resistant strains selected from the susceptible strain on artificial diets | | See Materials and methods, Laboratory strains |

*Continued on next page*

*Continued*

| Reagent type (species) or resource | Designation | Source or reference | Identifiers | Additional information |
|---|---|---|---|---|
| Strain, strain background (*H. armigera*) | Adult female bollworm moths | Collected from field | | See Materials and methods, Collection of field strains |
| Biological sample (*Helicoverpa armigera* densovirus-1) | HaDV2 | (*Xu et al., 2014*) DOI: 10.1371/journal.ppat.1004490 | | See Materials and methods, HaDV2 preparation |
| Recombinant DNA reagent | pEASY-T1 cloning vector | TransGen, Beijing, China | | See Materials and methods, Detection of HaDV2 in wild populations of *H. armigera* |
| Sequence-based reagent | HaDV-F | (*Xu et al., 2014*) DOI: 10.1371/journal.ppat.1004490 | | GGATTGGCCTG GGAAATGAC |
| Sequence-based reagent | HaDV-R | (*Xu et al., 2014*) DOI: 10.1371/journal.ppat.1004490 | | CGTTGTTTTTAT ATCCGAGG |
| Chemical compound, drug | Cry1Ac | Dow AgroSciences (Indianapolis, IN) | | See Materials and methods, Bt toxins |
| Software, algorithm | POLO Plus LeOra Software | POLO Plus LeOra Software, Berkeley | | Probit analysis |
| Software, algorithm | BLASTx | BLASTx | RRID:SCR_004870 | |
| Software, algorithm | Bowtie | Bowtie – 0.12.7 | RRID:SCR_005476 | |
| Software, algorithm | RSEM | RSEM – v1.1.17 | RRID:SCR_013027 | |
| Software, algorithm | DEseq2 | DEseq2 | RRID:SCR_015687 | |
| Software, algorithm | SPSS | SPSS | RRID:SCR_002865 | |
| Software, algorithm | RADR | (*An et al., 2015*) Doi: org/10.1002/ps.3807 | | The relative average development rates |

## Bt toxins

Cry1Ac was obtained as a gift from Dow AgroSciences (Indianapolis, IN) in product formulation MVPII (20%). To avoid degradation, Cry1Ac protoxin was stored at −80°C. To exclude the possibility of a decline in protoxin potency over time, the susceptible strains (LF) were used as the internal control in different years (*Cao et al., 2014*).

## Laboratory strains

Different *H. armigera* strains were used: two susceptible strains (LF and 96S) and eight Cry1Ac-resistant strains (BtR, 96CAD, LFC2, LF5, LF30, LF60, LF120, and LF240), with different resistance ratios due to different resistant mechanisms (*Liang et al., 2008*; *Xiao et al., 2014*). The Bt-susceptible strains, LF and 96S, were collected from Langfang, Hebei Province, in 1998, and Xinxiang, Henan Province, in 1996, respectively. They had been continuously cultured in the laboratory without exposure to Bt toxin. The resistant strains were selected from the susceptible strain on artificial diets (*Cao et al., 2014*; *Liang et al., 2008*); 96CAD (with a cadherin mutation) and LFC2 (with an ABCC2 mutation, unpublished) are two near-isogenic lines isolated from BtR and LF60 resistant strains, respectively (*Xiao et al., 2017*). All strains were reared on artificial diet. Rearing, selection, and bioassays were conducted at 25±1°C, photoperiod 14L:10D, and 75±10% relative humidity. To test whether the Bt could change the replication rate of HaDV2 in their hosts, newly hatched larvae from single pair matings, which both the parents and larvae were with low dose of HaDV2, fed on artificial

diets with either Bt (0.5 µg/g) or without Bt and quantified larval growth and virus titers 8 days later using the qPCR method (*Xu et al., 2014*).

## Collection of field strains

Adult female bollworm moths were collected from 1000 W light traps in two counties 360 km apart: Xiajin County, Shandong Province (with an intensive cotton planting area), and in Anci County, Hebei Province (with a multiple-crop farming system that includes cotton). The history of Bt-cotton deployment in the two locations during 2007–2013 has been reported previously (*An et al., 2015*). In the two counties, *H. armigera* female moths were trapped from June to October in all years, as previously reported (*An et al., 2015*).

## HaDV2 preparation

HaDV2-containing liquid fluids were prepared as described previously (*Xu et al., 2014*). Briefly, the collected individual moths were ground in liquid nitrogen completely. About 10 mg of powder were transferred to a clean 1.5 mL tube and DNA was extracted. PCR reactions were undertaken to detect the presence of HaDV2 using specific primers, HaDV-F: GGATTGGCCTGGGAAATGAC and HaDV-R: CGTTGTTTTTATATCCGAGG. The remaining debris of HaDV2-positive individuals was transferred to 1 mL phosphate-buffered saline buffer on ice. The homogenate was centrifuged and the liquid supernatant was subsequently filtered with 0.22 µm membrane filter (Sigma-Aldrich). The HaDV2-containing filtered liquid (200 µL per tube) was collected and stored immediately at −80°C. Quantification of the virus was performed using the qPCR method (*Xu et al., 2014*).

## Detection of HaDV2 in wild populations of *H. armigera*

Samples of larvae or adults were collected at 12 locations in 2014, 20 in 2015, and 18 in 2016 (*Supplementary file 1g*). Infection rates with HaDV-2 were determined using the PCR method described in *Xu et al., 2012* and the virus detection limits were tested with DNA extracted from HaDV2-positive individuals and the plasmid generated with the fragments amplified with the primers for detection of HaDV2 and pEASY-T1 cloning vector (TransGen, Beijing, China). This showed that the detection limit for HaDV2 with PCR method was >61.85 copies/µL or 0.13 ng/µL DNA (*Figure 6—figure supplement 2*).

## Bioassays

*H. armigera* strains were reared and assessed simultaneously for susceptibility to Cry1Ac, as previously reported (*Liang et al., 2008*). In brief, the susceptibility was evaluated by feeding *H. armigera* larvae with an artificial diet containing different concentrations of Cry1Ac toxin. Newly hatched larvae were first orally inoculated with either filtered-liquid containing HaDV2 or filtered-liquid from uninfected individuals. After that, they were placed in each treatment Petri dish for 2 days to ensure that larvae ingested the treated diet. They were then transferred individually on Cry1Ac-contaminated artificial diet in plastic wells (depth: 1.5 cm, vol. 3 mL) in 24-well plates. The plates were covered with a plastic lid to prevent escape and 72 larvae were tested per Cry1Ac concentration. After 7 days, the mortality was recorded and body mass of individuals that were still alive was measured. Both dead larvae and those with a body mass of less than 5 mg were recorded as dead. For each strain, the median lethal concentration ($LC_{50}$) value was determined using a Probit analysis (POLO Plus LeOra Software, Berkeley) and the resistance level to Cry1Ac indicated by the resistance ratio was calculated with $LC_{50}$ of the tested strains divided by the $LC_{50}$ of the susceptible strain.

## Assessment of fitness cost

To test the impact of HaDV2 infection on the fitness cost of its host, life-table parameters of the different strains were analyzed on artificial diet. The oral inoculation method was the same as previously described. Thirteen fitness components (life history parameters) were obtained with life-table techniques (*Supplementary file 1c and 1d*). The components considered were: (i) survival rate from the 1st to 5th larval instar (0–1), (ii) survival rate from the 5th instar to the pupal stage (0–1), (iii) duration of larval development (days), (iv) pupal weight (mg), (v) female pupal development period (days), (vi) male pupal development period (days), (vii) adult emergence rate (0–1), (viii) proportion of females (or sex ratio) (0–1), (ix) copulation rate (0–1), (x) female adult longevity (days), (xi) male

adult longevity (days), (xii) fecundity (number of eggs laid), and (xiii) hatching rate (0–1). The net reproductive rate ($R_0$) was calculated for each *H. armigera* strain (*Cao et al., 2014*) as $R_0=N_{t+1}/N_t$, where $N_t$ is the population size of the parent generation and $N_{t+1}$ is that of its next generation. When $R_0>1$, it indicates a higher number of female offspring produced than that of the parental females. We used 30 larvae per repeat and three repeats for each treatment.

To analyze the fitness cost associated with Cry1Ac-resistance, the larval weight of the different strains was analyzed on Bt/non-Bt-cotton. Newly hatched larvae were first orally inoculated with either filtered-liquid containing HaDV2 or filtered-liquid from non-infected individuals. After that, they were placed in each treatment Petri dish for 2 days to ensure that larvae ingested the treated diet. They were then transferred individually onto Bt-cotton (Zhongmian 29) or non-Bt-cotton (Zhongmian 24) leaf in plastic wells in 24-well plates. After 9 days, the weight of individuals was measured. Each treatment had 48 larvae with three repeats.

## Bioassay of F1 generation on Bt and non-Bt diets

To obtain F1 generation offspring of each of the laboratory strains, one virgin adult male and one virgin adult female were paired in a 250 mL clear plastic cup. Adult females collected from the field were placed individually into 250 mL clear plastic cups and covered with gauze to provide a substrate for egg laying. Eggs were collected on a daily basis. At larval hatch, 24 neonates from each female line were placed on non-Bt diet, and 24 neonates were placed on Cry1Ac-containing diet (1 μg of Cry1Ac/g diet). The concentration of Bt Cry1Ac protoxin was 1.0 μg/g diet. The composition of the diet was previously described (*Chen et al., 2015*). Rearing, selection, and bioassays were conducted at 27±2°C with a photoperiod of 14:10 h L:D and a relative humidity of 75±10%. Larvae were scored for the developmental stage after 6 days. Larval instar was determined on the basis of head capsule and body size.

## Analyzing the effects of the HaDV2 on its hosts by transcriptome analysis

To determine the effect of HaDV2 on *H. armigera* at a transcriptomic level, we collected samples of the HaDV2-negative and HaDV2-positive individuals from single pairs of *H. armigera* and performed RNA-seq, using larvae at 24, 48, and 72 hr after hatching. Briefly, newly hatching larvae were fed on filtered liquid with/without HaDV2 as described previously (*Xu et al., 2014*). Samples were collected at 24, 48, and 72 hr and there were two groups of 30 individuals for each group per trial (*Supplementary file 1h*). The Trinity (v2.0.6) software (*Grabherr et al., 2011*) was used to assemble the clean reads with default parameters. BLASTx was performed to align the assembled contigs from Trinity to the database of NR, String, Swissprot, and KEGG for functional annotation. The e-value cutoff was set at 1E−5 for further analysis. For gene expression analysis, the reads from 12 libraries were mapped to the assembled contigs using Bowtie 0.12.7 (*Langmead et al., 2009*) with no more than two mismatches within the first 28 bp. The read counts accumulated on the contigs were normalized as fragments per kilobase of exon model per million mapped reads (FPKM) values (*Trapnell et al., 2010*). Quantitative analysis for each unigene was estimated using FPKM values by RSEM (v1.1.17) software (*Li and Dewey, 2011*) with default parameters. Then the R package DEseq2 (*Love et al., 2014*) was used to get the significantly differential expressed unigenes at different comparisons. The threshold to determine the significantly differential expressed unigenes was 'fold change≥1.5 and the p<0.05'. The hierarchical clustering method was applied to analyze the expression pattern of significantly differentially expressed unigenes in different samples. Significantly enriched KEGG pathways were identified using the Fisher's exact test (p<0.05) (*Klopfenstein et al., 2018*).

## RNA sequence analysis

Offspring from a single uninfected breeding pair were reared to produce the N-strain (uninfected) laboratory culture. Neonate N-strain larvae were first orally inoculated with either filtered-liquid containing HaDV2 ($10^8$ copies/μL) or filtered-liquid from non-infected individuals (control). One hundred N-strain neonates were placed in each treatment Petri-dish for 2 days to ensure that larvae ingested the treated diet. They were then transferred to a 24-well plate (one individual per well: diameter=1.5 cm; height=2 cm) containing the artificial diet containing 1 μg/mL Cry1Ac toxins. After 48 hr, the

larvae were collected and stored at −80℃ for transcriptome sequencing. Twenty larvae were pooled together as a sample. A total of 12 samples were sequenced including three replications of four treatments (HaDV2-Bt−, HaDV2−Bt+, HaDV2+Bt−, and HaDV2+Bt+).

## Quantification and statistical analysis

The RADRs were calculated according to *An et al., 2015*. In brief, the RADR for a line was calculated as the average body length rating of larvae from that line reared on the Bt diet divided by the average rating of larvae from that line reared on the non-Bt diet. Multivariate two-factor variance analysis was conducted with IBM SPSS 20. Student's t-test or ANOVA with Tukey-test post hoc comparisons were used to determine the level of significance. Simple linear regression model was used to analyze the temporal trends of *H. armigera*, including RADR and HaDV2 infection rate, from 2007 to 2016. Simple linear regression model was also used to analyze the association relationship of RADR and HaDV2 infection rate in different years. Logistic regression, using the R statistical package (version 3.3.3) (*R Development Core Team, 2017*), was used to test the association between HaDV2 infection rate and environmental variables (rainfall and temperature) and altitude the proportional cover of the main crops (cotton, rice, corn, wheat, beans, tubers, oil-producing crops, and vegetables), the Bt status of the cotton crop, and the number of years since the introduction of Bt-cotton to a province (defined as the year in which Bt-cotton comprised at least 10% of the cotton grown in a province).

## Acknowledgements

The authors thank Bruce E Tabashnik (Department of Entomology, University of Arizona, USA) and Ling Wang (Institute of Plant Protection and Soil Fertility, Hubei Academy of Agricultural Sciences, China) for their comments on data analysis.

## Additional information

### Competing interests

Mario Soberón, Alejandra Bravo: coauthor of a patent on modified Bt toxins, "Suppression of Resistance in Insects to Bacillus thuringiensis Cry Toxins, Using Toxins that do not Require the Cadherin Receptor" (patent numbers: CA2690188A1, CN101730712A, EP2184293A2, EP2184293A4, EP2184293B1, WO2008150150A2, WO2008150150A3). The other authors declare that no competing interests exist.

### Funding

| Funder | Grant reference number | Author |
| --- | --- | --- |
| The Key Project for Breeding Genetic Modified Organisms | 2019ZX08012004 | Kongming Wu |
| The Key Project for Breeding Genetic Modified Organisms | 2016ZX08012004 | Kongming Wu |
| Biotechnology and Biological Sciences Research Council | BB/L026821/1 | Kenneth Wilson |
| Research Councils UK | BB/P023444/1 | Kenneth Wilson |

The funders had no role in study design, data collection and interpretation, or the decision to submit the work for publication.

### Author contributions

Yutao Xiao, Formal analysis, Validation, Methodology; Wenjing Li, Data curation, Investigation; Xianming Yang, Resources, Formal analysis, Supervision; Pengjun Xu, Data curation, Formal analysis, Validation, Investigation, Methodology, Writing - original draft, Writing - review and editing; Minghui Jin, Software, Methodology; He Yuan, Software, Methodology, Writing - original draft; Weigang Zheng, Data curation, Software; Mario Soberón, Alejandra Bravo, Resources, Methodology, Writing

- original draft; Kenneth Wilson, Data curation, Software, Funding acquisition, Methodology, Writing - original draft, Project administration, Writing - review and editing; Kongming Wu, Resources, Supervision, Funding acquisition, Validation, Methodology, Writing - original draft, Project administration, Writing - review and editing

### Author ORCIDs
Kenneth Wilson ⬤ http://orcid.org/0000-0001-5264-6522
Kongming Wu ⬤ https://orcid.org/0000-0003-3555-4292

### Decision letter and Author response
Decision letter https://doi.org/10.7554/eLife.66913.sa1
Author response https://doi.org/10.7554/eLife.66913.sa2

## Additional files

### Supplementary files
• Supplementary file 1. Supplementary tables. (**a**) Bt toxin sensitivity test of different *Helicoverpa armigera* strains with or without HaDV2 infection. (**b**) Mortality changes with Bt toxin concentration test of different *H. armigera* strains with or without HaDV2 infection. (**c**) Comparing the effects of HaDV2 on fitness components of LF, LF5, LF60, and LF240. LF is susceptible strain; LF5, LF60, and LF240 are Bt resistant strains selected with LF strain. Significant differences (ANOVA followed by Tukey's HSD test) between each strain with or without HaDV2 infestation are indicated by different letters (p<0.05). Insects were reared on artificial non-Bt-diet. D+ stand for infected by HaDV2, D− stand for non-infected by HaDV2. (1–5) means the survival rate from the first star to the 5th star; 5 p: from the 5th star to pupa; Proportion FA: the rate of female divided male. (**d**) Analysis of variance for fitness parameters of 4 cotton bollworm strains (LF, LF5, LF60, and LF240). (**e**) Fitness of the susceptible and resistant strains on Bt- and non-Bt-cotton infected with or without HaDV2. (**f**) Analysis of variance for weight of cotton bollworm larvae. (**g**) Sample information and infection rates of HaDV2 in the field populations of *H. armigera*. See *Figure 6* for a map of locations. X: east longitude, Y: northern latitude. (**h**) Host information and accessions for samples.

• Transparent reporting form

### Data availability
RNA-seq data have been deposited in NCBI, see Supplementary file 8. All data generated or analysed during this study are included in the manuscript and supplementary file.

The following datasets were generated:

| Author(s) | Year | Dataset title | Dataset URL | Database and Identifier |
|---|---|---|---|---|
| Xiao YT, Li WJ, Yang XM, Xu PJ, Jin MH, Yuan H, Zheng WG, Soberón M, Bravo A, Wilson K, Wu KM | 2021 | Rapid spread of a symbiotic virus in a major crop pest following wide-scale adoption of Bt-cotton in China | https://www.ncbi.nlm.nih.gov/bioproject/?term=PRJNA638220 | NCBI BioProject, PRJNA638220 |
| Xiao YT, Li WJ, Yang XM, Xu PJ, Jin MH, Yuan H, Zheng WG, Soberón M, Bravo A, Wilson K, Wu KM | 2021 | Rapid spread of a symbiotic virus in a major crop pest following wide-scale adoption of Bt-cotton in China | https://www.ncbi.nlm.nih.gov/bioproject/?term=PRJNA638972 | NCBI BioProject, PRJNA638972 |

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
