## [Decision Letter]

**Acceptance summary:**

Your work, which will be of interest to a broad audience interested in microbe-insect interactions and how they may affect adaptation to pesticides, nicely indicates that infection with a mutualistic virus can enhances fitness of a moth, and that selection pressure represented by transgenic crops may be driving the spread of mutualistic infections in moth populations.

**Decision letter after peer review:**

Thank you for submitting your article "Rapid spread of a symbiotic virus in a major crop pest following wide-scale adoption of Bt-cotton in China" for consideration by *eLife*. Your article has been reviewed by 2 peer reviewers, and the evaluation has been overseen by a Reviewing Editor and Detlef Weigel as the Senior Editor. The reviewers have opted to remain anonymous.

Essential Revisions:

1) Change the title to more precisely reflect the most certain conclusions of the study.

2) Clearly distinguish between resistance (the evolution of increased tolerance due to genetic changes in the pest population) and other mechanisms of increased tolerance (e.g. virus infection).

3) Provide the minimum detection limit for the densovirus, to ensure that latent infection is not missed.

4) In the Discussion, consider and evaluate an alternative hypothesis; that increased virus infection and increased tolerance of Bt cotton are parallel but independent responses to the stress of feeding on Bt cotton.

5) Redraw Figure 1 with the more appropriate units of LC50 rather than resistance ratio.

6) Clarify questions about the identity and uniqueness of the virus.

7) In the Discussion, use existing data to estimate the potential impact on reduction of pest control efficacy by Bt cotton due to densovirus infection in relevant cotton-growing areas of China; and compare this impact to estimated reduction in efficacy due to cadherin mutations and due to mutations in the ABC transporter.

8) In the Discussion, describe the ideal approach/dataset that would conclusively show that infection by the densovirus causes (not merely is associated with) the increased tolerance of Bt cotton, and evaluate this approach with respect to feasibility. Likewise evaluate the approaches presented in the manuscript relative to this ideal approach.

9) In the Discussion, discuss the feasibility or infeasibility of manipulating the virus infection status to increase the pest control efficacy by Bt cotton.

*Reviewer #1:*

Previous reports have provided evidence identifying infection of cotton bollworm with a densovirus as resulting in increased fitness. In the current manuscript, the relevance of this infection towards field resistance to transgenic Bt corn is evaluated by comparing its incidence between regions in China growing non-Bt versus Bt cotton. A clear correlation emerges with infection rates being higher in Bt versus non-Bt cotton growing areas, although its effect on resistance to Cry1Ac and Bt cotton is not as clear.

Strengths:

The manuscript presents evidence for the spread of densovirus infection in field bollworm populations, and that this spread seems to occur at a faster rate in areas of China where Bt cotton is grown versus non-Bt cotton areas.

Life table comparisons clearly show increased fitness in bollworms infected with the virus.

The study capitalizes on availability of an impressive collection of samples with distinct geographic and historic origin to address relevant evolutionary questions.

Weaknesses:

The suggested role for densovirus infection in resistance to Cry1Ac and Bt cotton is supported by association and the data presented does not necessarily support causation. In fact, the confidence intervals in all the comparisons from bioassays overlap substantially and the resulting resistance ratio is not a good estimate of any significant differences the infection may have on ability to survive Cry1Ac. Infection by a virus is expected to activate the immune system, so the larvae used in bioassays should be considered as "primed" and the slight reduction in susceptibility should not be considered as an effect of the virus itself.

The life table data clearly shows that fertility and fecundity are probably the most relevant aspects affecting fitness of infected insects. These differences in reproduction (even more than differences in larval growth) could explain why infection is rapidly spreading in the wild. However, most of the research and analyses are focused on the possibility that the viral infection may make the insects more able to survive Cry1Ac or Bt cotton. There are no conclusive data supporting this hypothesis in the current version, other than increased infection rates in Bt-cotton growing areas. This could be explained by effects on reproduction rather than enhanced survival.

Related to this aspect, there should be a more clear distinction between the densovirus increasing fitness versus increasing resistance, the data supports the former but is not so clear in the later.

It would be useful to provide a map detailing regions were moths were collected.

The putative effect of infection on susceptibility could be strengthened by comparing effects of treating the larvae with a non-pathogenic microbe other than the densovirus. One would expect a similar priming of the immune response, making the larva less susceptible to subsequent exposure to Cry1Ac.

The text may be misleading as written as in several statements suggests densovirus infection is related to resistance to Cry1Ac.Resistance is not due to the densovirus infection, but to other mechanisms.

More focus needs to be put on the effect of viral infection increasing reproductive output rather than larval growth. It is plausible that the former will have a more relevant role on the spread of infection. The observed higher frequency of infection in Bt cotton areas could be due to participation of infection in survival (as hypothesized) or may be a result of initial selection of individuals that were infected and subsequent spread due to higher reproductive output.

*Reviewer #2:*

The various pathogenic, parasitic, symbiotic, and mutualistic interactions between insects and the microbes they interact with represents a rich area of research. This study by Xiao et al. represents a very interesting example of such a relationship. Overall the study is well designed and executed. The approach they utilize to test their hypothesis is valid and they combined both laboratory and field collected insects to address the question. The RNAseq analysis also provides potential insights into possible mechanisms by which the virus HaDV2 enables enhanced resistance to Bt Cry1Ac. The RNAseq data also represent one of the minor issues. The authors focused on analyzing only development and immune systems, however, they do not report on any other significantly different changes in gene expression other than reporting that there were 1573 significant differences. The authors should at least provide some holistic analysis and report the data in the supplemental results. Focusing on development and immune systems is valid and rationally supported but a complete analysis should be presented. The relationship between *H. armigera* and HaDV2 is more a mutualistic relationship, thus, the authors should consider changing the titles of the manuscript and the supplementary data. This is an exciting study and is well written and will be of general interest to the field.

In this study Xiao et al., report on the association of an insect virus with the lepidopteran host *Heliothis armigera* and the development of increased resistance to the Bt toxin Cry1Ac in transgenic cotton. Previous studies from this group had shown that infection of *H. armigera* with the densovirus *H. armigera* densovirus^-1^ (HaDV2) enhanced resistance to baculovirus infections and potentially increased resistance to Cry1Ac with no detrimental impacts to fitness. In this study they test the hypothesis that the large adoption of transgenic Bt-cotton has selected for *H. armigera* infected with HaDV2. To test this they examined different *H. armigera* strains with and without HaDV2 and observed a significant increase in resistance in all strains. Previous studies has shown Bt resistance has fitness costs, they showed that resistant strains had improved reproductive rate if the strain was infected with HaDV2 apparently compensating for the fitness costs of resistance. Analysis of field collected larvae from Bt-cotton and non-Bt-cotton regions over the last 10 years showed a direct correlation with HaDV2 infection. To address the question of molecular basis for the enhanced resistance to Bt cotton when infected with HaDV2, resistant and non-resistant strains with or without HaDV2 infection were analysed by RNAseq. Differences in gene expression was bioinformatically analyzed. Focusing on biochemical pathways related to development and immune systems a number of pathways were identified which could account for the observed enhanced fitness of Bt resistant larvae. They conclude the selection of the this mutualistic virus has been selected for by the widespread use of Bt-cotton.

Additional guidance for Essential Revisions:

1) Title: The word "symbiotic" should be removed from the title. Since viruses are not free-living, they are symbiotic by definition (see for example the definitions in Roossinck 2011, which is cited). Whether the word "mutualistic" can be used in its place will depend on whether the authors can demonstrate a causal link between presence of the virus and greater fitness (not sufficiently shown in the current version). "Densovirus" should replace "virus" in the title.

2) Use of the word "resistance" must be confined to genetic changes in a population which increase its tolerance to an external stress as a result of previous exposure to that stress. "Resistance" cannot be used as a synonym for "tolerance". There is no evidence presented in the manuscript for *H. armigera* resistance to the granulovirus. When discussing the response of the insect to the virus, words like "tolerance", "virulence", or "susceptibility" must be used. If an individual insect's tolerance to virus infection increases as a result of feeding on Bt cotton, this is not resistance. Previous publications from this group also misused the word "resistance", but that is not a sufficient reason for its continued misuse here.

By the same token, any increase in an individual insect's tolerance to Bt cotton caused by exposure to the virus, or genes upregulated in response to the virus, is not resistance, and must not be referred to as such. "Enhanced resistance" or "increase in resistance level" may not be used. Instead, two components of tolerance should be discussed, one due to genetic resistance to Cry toxin caused by a specific mechanism (e.g. mutations in the cadherin or ABC transporter), and the other due to the difference between presence or absence of the virus.

3) Detection limit for latent virus: Has the virus actually spread, or has the stress of Bt cotton converted it from a universally-present latent form to an active form in some stressed individuals? A latent virus might be present in all individuals--can the authors rule out this possibility? The authors need to give the detection limit of their PCR technique, in term of virus genomes per single-copy gene per cell of the insect. The reference cited (Xu et al. 2014 PLoS Pathogens) describes the use of a standard curve for copy number estimation but does not describe the detection limit. The criterion for when a field-collected insect is declared *not* to harbor the virus must be explicitly stated here.

4) Parallel response to stress: Apart from the issue of latent viruses; the higher incidence of virus in areas of Bt cotton than non Bt-cotton, and the increase in virus incidence with the amount and proportion of Bt-cotton over time, could be partly or completely explained by increased stress on the insects feeding on Bt cotton, that weakens their defenses against the virus and allows the virus to increase within the insect. How can the authors rule this out? Even if this is only a partial explanation, how much of the higher incidence of virus could be explained by this factor, rather than a protective role of the virus against Bt toxin?

5) In Figure 1, the slope is dependent on which strain is chosen for the susceptible reference, and the labelling of the axes confounds the two components of tolerance. The figure should be redrawn with LC50 as the units on both axes. The scale may be either linear or logarithmic.

6) Uniqueness of the virus: Two abbreviations for the virus are used: HaDNV-1 and HaDV2; mostly HaDV2. The study by Xu et al. uses HaDNV-1. The GenBank record (HQ613271, cited by Xu et al. 2014) calls it "*Helicoverpa armigera* densovirus strain SDBH2010-1". Please provide enough information in the Supplement to remove any doubt about which virus is being discussed. Since the GenBank record names a specific strain, can we assume that there are many different strains infecting *H. armigera* in nature? This is also implied by the switch from -1 to -2 in the abbreviation. If there is an HaDV2, does that mean there is also a HaDV1? Which of them can be detected by the PCR method? Which of them are being counted in the increase?

7) Estimation of potential impact: This will require assumptions about the relationship between the reproductive rate and crop damage caused by the pest. Clearly state these assumptions, possibly using two or three different relationships, so that their robustness can be evaluated.

8) Ideal demonstration of causality: In the laboratory, this would involve infecting a virus-free strain of bollworm, and then curing the strain of the virus so that pre-infection, infection, and post-infection performance could be compared. Is this even possible in the laboratory? In the field?

9) Evidence was presented in earlier work that densovirus infection can reduce baculovirus infection. Does a latent or active baculovirus infection have any prospects to reverse the effect of densovirus on tolerance of Bt-cotton?

---

## [Author Response]

Essential Revisions:1) Change the title to more precisely reflect the most certain conclusions of the study.

We changed the title to “Rapid spread of a densovirus in a major crop pest following wide-scale adoption of Bt-cotton in China”.

2) Clearly distinguish between resistance (the evolution of increased tolerance due to genetic changes in the pest population) and other mechanisms of increased tolerance (e.g. virus infection).

We have adjusted several expressions in the article in order to better distinguish resistance and tolerance. (page 2, lines 31, 34; page 3, lines 56, 58, 71, 73; page 4 , lines 79, 87, 89, 93, 97, 98; page 5, lines 100, 102, 106, 109; page 6, line 137; page 9, line 209; page 10, line 239; page 11, line 261; page 12, line 280; page 16, line 383; page 17, line 422;page 18, line 426; page 19, line 473; page 20, lines 475, 477; page 21, lines 510).

3) Provide the minimum detection limit for the densovirus, to ensure that latent infection is not missed.

Accepted, we added the detection limit of PCR for HaDV2.

“the virus detection limits were tested with DNA extracted from HaDV2-positive individuals and the plasmid generated with the fragments amplified with the primers for detection of HaDV2 and pEASY-T1 cloning vector (TransGen, Beijing, China). This showed that the detection limit for HaDV2 with PCR method was > 61.85 copies/µL or 0.13 ng /µL DNA (Figure 6 —figure supplement 2).” (page 26, lines 569-574).

4) In the Discussion, consider and evaluate an alternative hypothesis; that increased virus infection and increased tolerance of Bt cotton are parallel but independent responses to the stress of feeding on Bt cotton.

Accepted, we provided additional experiment and added some discussion about the relationship between HaDV2 infection and the tolerance of Bt cotton as follows:

Methods: To test whether the Bt could change the replication rate of HaDV2 in their hosts, newly-hatched larvae from single pair matings, which both the parents and larvae were with low dose of HaDV2, fed on artificial diets with either Bt (0.5 µg/g) or without Bt and quantified larval growth and virus titres 8 days later using the qPCR method (Xu et al., 2014). (page 25, lines 538-542)

Results: At 8 days post-hatching, *H. armigera* larvae were significantly lighter (t = 10.164, d.f. = 32, P < 0.0001, n=17) and had lower HaDV2 viral loads in individuals feeding on diet containing Bt than the ones without Bt (t = 4.527, d.f. = 32, P < 0.0001, n=17), suggesting that Bt decreased the replication rate of HaDV2 by suppressing the growth of *H. armigera* larvae (Figure 1 —figure supplement 1). (page 5, lines 107-112).

Discussion: Using the PCR method, it is not possible to discount the possibility that HaDV2 was widespread in field populations of *H. armigera* as very low-titre covert infections, and that the higher incidence of virus in areas of Bt-cotton was due to the stress associated with larvae feeding on plants expressing Bt toxin. To test the hypothesis, using larvae with low dose of HaDV2 from single pair, we quantified the copy numbers of HaDV2 within the cotton bollworms feeding on artificial diets with or without Bt toxin. Interestingly, the results showed that feeding on the Bt toxin resulted in lower titres of HaDV2, which might be due to the Bt suppressing the growth of *H. armigera* larvae. These results suggest that the higher infection rate of HaDV2 might be due to lower mortality by higher Bt tolerance and the increased reproduction of the cotton bollworms with HaDV2 than those without. (page 15, lines 352-362).

5) Redraw Figure 1 with the more appropriate units of LC50 rather than resistance ratio.

Accepted. The figure 1 were redrew by LC50 of *H. armigera* strains with or without HaDV2 infection. (page 36, lines 940-945)

6) Clarify questions about the identity and uniqueness of the virus.

Accepted (Page 3, Line 54). HaDNV-1 and HaDV2 are pseudonyms for the same virus we found in *Helicoverpa armigera*. At the beginning, densovirus was abbreviated as “DNV” (Tijssen, et al., 1976). For distinguishing the densovirus with ambisense genome in *H. armigera* which was named as HaDNV (El-Far, et al., 2012), we named the densovirus we find as HaDNV-1 (Xu, et al., 2012; Xu, et al., 2014). However, the ICTV changed the abbreviation of densovirus as “DV” and named the HaDNV as HaDV1 in 2014 (Cotmore, et al., 2014). Accordingly, we changed HaDNV-1 to HaDV2 and using HaDV2 stand for the densovirus we found in newly published paper (Xu, et al., 2017a; Xu, et al., 2017b). We added the illustration (page 3, lines 54-55).

Cotmore SF, Agbandje-McKenna M, Chiorini JA, Mukha DV, Pintel DJ, Qiu J, Soderlund-Venermo M, Tattersall P, Tijssen P, Gatherer D, Davison AJ 2014. The family Parvoviridae. Arch Virol 159: 1239-1247. doi: 10.1007/s00705-013-1914-1

El-Far M, Szelei J, Yu Q, Fediere G, Bergoin M, Tijssen P 2012. Organization of the Ambisense Genome of the Helicoverpa armigera Densovirus. J Virol 86: 7024.

Tijssen P, van den Hurk J, Kurstak E 1976. Biochemical, biophysical, and biological properties of densonucleosis virus. I. Structural proteins. J Virol 17: 686-691.

Xu P, Cheng P, Liu Z, Li Y, Murphy RW, Wu K 2012. Complete genome sequence of a monosense densovirus infecting the cotton bollworm, Helicoverpa armigera. J Virol 86: 10909.

Xu P, Graham RI, Wilson K, Wu K 2017a. Structure and transcription of the Helicoverpa armigera densovirus (HaDV2) genome and its expression strategy in LD652 cells. Virol J 14: 23.

Xu P, Liu Y, Graham RI, Wilson K, Wu K 2014. Densovirus Is a Mutualistic Symbiont of a Global Crop Pest (Helicoverpa armigera) and Protects against a Baculovirus and Bt Biopesticide. PLoS Pathog 10: e1004490.

Xu P, Yuan H, Yang X, Graham RI, Liu K, Wu K 2017b. Structural proteins of Helicoverpa armigera densovirus 2 enhance transcription of viral genes through transactivation. Arch Virol 162: 1745-1750.

7) In the Discussion, use existing data to estimate the potential impact on reduction of pest control efficacy by Bt cotton due to densovirus infection in relevant cotton-growing areas of China; and compare this impact to estimated reduction in efficacy due to cadherin mutations and due to mutations in the ABC transporter.

Accepted. We added to the discussion. The Bt resistance induced by ABCC2 and cadherin is due to genomic heritable variation (Liang et al., 2008; Xiao et al., 2014), while the Bt tolerance induced by HaDV2 appears to be due to the regulation of immune and other pathways. ABCC2 and cadherin can directly mediate high levels of resistance to Bt toxin in *H. armigera* but with significant fitness costs (e.g. lower reproduction) (Cao et al., 2014), however, HaDV2 infection could enhance host reproduction and tolerance to Cry1Ac in both susceptible and resistant strains of *H. armigera*, increasing larval survival rate. (pages 16-17, lines 394-401).

8) In the Discussion, describe the ideal approach/dataset that would conclusively show that infection by the densovirus causes (not merely is associated with) the increased tolerance of Bt cotton, and evaluate this approach with respect to feasibility. Likewise evaluate the approaches presented in the manuscript relative to this ideal approach.

We have added a new section to the Discussion:

“Whilst data from the field are inevitably correlational, when combined with the laboratory experiments, we feel that the most parsimonious explanation for the observations reported here is that HaDV2 directly enhances tolerance of *H. armigera* to Bt in the field, probably via increasing larval growth rates and via priming of immune defences. Demonstrating causation conclusively in the field will prove difficult however. One possibility would be to plant some Bt cotton in areas that are currently Bt-free and to compare larval growth patterns and HaDV2 infection levels in neighbouring Bt and non-Bt areas, mirroring the ‘natural’ field experiments we report here. Even this approach, though, would not demonstrate causality conclusively. We envisage that HaDV2 infection levels will eventually go to fixation (or close to it) in both Bt- and non-Bt areas, due to the growth benefits associated harbouring this mutualist but that the time to get to this point will take longer in non-Bt areas, due to the lower fitness benefits. Likewise, we envisage that the spread of HaDV2 infection would slow down in Bt-areas if Bt-cotton were to be replaced by conventional cotton or a GM variety that did not rely on Bt toxins.” (pages 20, lines 470-493).

9) In the Discussion, discuss the feasibility or infeasibility of manipulating the virus infection status to increase the pest control efficacy by Bt cotton.

Accepted, we added the discussion as follows: “Although we show that HaDV2 increases the tolerance of its host to Bt, it is not feasible to manipulate HaDV2 infection status in the field in order to enhance the capacity of Bt to manage *H. armigera* because of the lack of suitable anti-virals (against HaDV2) and because of the efficient vertical and horizontal transmission of this virius in the field.” (pages 21, lines 512-516).

Reviewer #1:Previous reports have provided evidence identifying infection of cotton bollworm with a densovirus as resulting in increased fitness. In the current manuscript, the relevance of this infection towards field resistance to transgenic Bt corn is evaluated by comparing its incidence between regions in China growing non-Bt versus Bt cotton. A clear correlation emerges with infection rates being higher in Bt versus non-Bt cotton growing areas, although its effect on resistance to Cry1Ac and Bt cotton is not as clear.Strengths:The manuscript presents evidence for the spread of densovirus infection in field bollworm populations, and that this spread seems to occur at a faster rate in areas of China where Bt cotton is grown versus non-Bt cotton areas.Life table comparisons clearly show increased fitness in bollworms infected with the virus.The study capitalizes on availability of an impressive collection of samples with distinct geographic and historic origin to address relevant evolutionary questions.Weaknesses:The suggested role for densovirus infection in resistance to Cry1Ac and Bt cotton is supported by association and the data presented does not necessarily support causation. In fact, the confidence intervals in all the comparisons from bioassays overlap substantially and the resulting resistance ratio is not a good estimate of any significant differences the infection may have on ability to survive Cry1Ac. Infection by a virus is expected to activate the immune system, so the larvae used in bioassays should be considered as "primed" and the slight reduction in susceptibility should not be considered as an effect of the virus itself.The life table data clearly shows that fertility and fecundity are probably the most relevant aspects affecting fitness of infected insects. These differences in reproduction (even more than differences in larval growth) could explain why infection is rapidly spreading in the wild. However, most of the research and analyses are focused on the possibility that the viral infection may make the insects more able to survive Cry1Ac or Bt cotton. There are no conclusive data supporting this hypothesis in the current version, other than increased infection rates in Bt-cotton growing areas. This could be explained by effects on reproduction rather than enhanced survival.Related to this aspect, there should be a more clear distinction between the densovirus increasing fitness versus increasing resistance, the data supports the former but is not so clear in the later.It would be useful to provide a map detailing regions were moths were collected.The putative effect of infection on susceptibility could be strengthened by comparing effects of treating the larvae with a non-pathogenic microbe other than the densovirus. One would expect a similar priming of the immune response, making the larva less suceptible to subsequent exposure to Cry1Ac.

Many thanks for the suggestions. That might be interesting and we also tried to construct strains of *H. armigera* with *Wolbachia* but we failed due to very low infection rate of *Wolbachia* in field populations of *H. armigera*. We didn’t find any other non-pathogenic microbes in field populations of *H. armigera* but if we do, the proposed experiment would be feasible.

The text may be misleading as written as in several statements suggests densovirus infection is related to resistance to Cry1Ac.Resistance is not due to the densovirus infection, but to other mechanisms.

Accepted. HaDV2 infection could increases the tolerance level of *H. armigera* to Cry1Ac toxin. But Resistance is not due to the HaDV2 infection. In order to avoid misleading readers, we have made necessary revisions to the relevant descriptions in the results and Discussion section of manuscript (e.g. adjusted several expressions to better distinguish resistance and tolerance). Please see details in the revised manuscript with tracking changes.

More focus needs to be put on the effect of viral infection increasing reproductive output rather than larval growth. It is plausible that the former will have a more relevant role on the spread of infection. The observed higher frequency of infection in Bt cotton areas could be due to participation of infection in survival (as hypothesized) or may be a result of initial selection of individuals that were infected and subsequent spread due to higher reproductive output.

We cannot be sure about the relative importance of increased growth, survival or reproduction in determining the spread of HaDV2 in *H. armigera*, but it is worth noting that the reproductive rate of HaDV2-positive individuals is ~20% greater than that of HaDV2-negative individuals and that survival from egg to adult is also ~20% higher for individuals carrying HaDV2, suggesting that both life-history traits contribute to the enhanced *R_0_* of HaDV2-infected individuals.

Additional guidance for Essential Revisions:1) Title: The word "symbiotic" should be removed from the title. Since viruses are not free-living, they are symbiotic by definition (see for example the definitions in Roossinck 2011, which is cited). Whether the word "mutualistic" can be used in its place will depend on whether the authors can demonstrate a causal link between presence of the virus and greater fitness (not sufficiently shown in the current version). "Densovirus" should replace "virus" in the title.

Accepted. We changed the title to “Rapid spread of a densovirus in a major crop pest following wide-scale adoption of Bt-cotton in China”.

2) Use of the word "resistance" must be confined to genetic changes in a population which increase its tolerance to an external stress as a result of previous exposure to that stress. "Resistance" cannot be used as a synonym for "tolerance". There is no evidence presented in the manuscript for H. armigera resistance to the granulovirus. When discussing the response of the insect to the virus, words like "tolerance", "virulence", or "susceptibility" must be used. If an individual insect's tolerance to virus infection increases as a result of feeding on Bt cotton, this is not resistance. Previous publications from this group also misused the word "resistance", but that is not a sufficient reason for its continued misuse here.By the same token, any increase in an individual insect's tolerance to Bt cotton caused by exposure to the virus, or genes upregulated in response to the virus, is not resistance, and must not be referred to as such. "Enhanced resistance" or "increase in resistance level" may not be used. Instead, two components of tolerance should be discussed, one due to genetic resistance to Cry toxin caused by a specific mechanism (e.g. mutations in the cadherin or ABC transporter), and the other due to the difference between presence or absence of the virus.

Accepted. We changed “resistance” to “tolerance” and changed several descriptions in this paper in order to better distinguish resistance and tolerance (page 2, lines 31, 34; page 3, lines 56, 58, 71, 73; page 4 , lines 79, 87, 89, 93, 97, 98; page 5, lines 100, 102, 106, 109; page 6, line 137; page 9, line 209; page 10, line 239; page 11, line 261; page 12, line 280; page 16, line 383; page 17, line 422;page 18, line 426; page 19, line 473; page 20, lines 475, 477; page 21, lines 510).

3) Detection limit for latent virus: Has the virus actually spread, or has the stress of Bt cotton converted it from a universally-present latent form to an active form in some stressed individuals? A latent virus might be present in all individuals--can the authors rule out this possibility? The authors need to give the detection limit of their PCR technique, in term of virus genomes per single-copy gene per cell of the insect. The reference cited (Xu et al. 2014 PLoS Pathogens) describes the use of a standard curve for copy number estimation but does not describe the detection limit. The criterion for when a field-collected insect is declared not to harbor the virus must be explicitly stated here.

Accepted, please see the answer for Essential Revisions Question 3.

4) Parallel response to stress: Apart from the issue of latent viruses; the higher incidence of virus in areas of Bt cotton than non Bt-cotton, and the increase in virus incidence with the amount and proportion of Bt-cotton over time, could be partly or completely explained by increased stress on the insects feeding on Bt cotton, that weakens their defenses against the virus and allows the virus to increase within the insect. How can the authors rule this out? Even if this is only a partial explanation, how much of the higher incidence of virus could be explained by this factor, rather than a protective role of the virus against Bt toxin?

Accepted, Accepted, please see the answer for Essential Revisions Question 4.

5) In Figure 1, the slope is dependent on which strain is chosen for the susceptible reference, and the labelling of the axes confounds the two components of tolerance. The figure should be redrawn with LC50 as the units on both axes. The scale may be either linear or logarithmic.

Please see the answer for Essential Revisions Question 5.

6) Uniqueness of the virus: Two abbreviations for the virus are used: HaDNV-1 and HaDV2; mostly HaDV2. The study by Xu et al., uses HaDNV-1. The GenBank record (HQ613271, cited by Xu et al. 2014) calls it "Helicoverpa armigera densovirus strain SDBH2010-1". Please provide enough information in the Supplement to remove any doubt about which virus is being discussed. Since the GenBank record names a specific strain, can we assume that there are many different strains infecting H. armigera in nature? This is also implied by the switch from -1 to -2 in the abbreviation. If there is an HaDV2, does that mean there is also a HaDV1? Which of them can be detected by the PCR method? Which of them are being counted in the increase?

Accepted, please see the answer for Essential Revisions Question 6.

7) Estimation of potential impact: This will require assumptions about the relationship between the reproductive rate and crop damage caused by the pest. Clearly state these assumptions, possibly using two or three different relationships, so that their robustness can be evaluated.

Estimating the potential impact of HaDV2 and its interaction with Bt-cotton would probably require an extensive agronomic modelling exercise, which is beyond the scope of the present study, but it is worth noting that the estimated fundamental net reproductive rates, R_0_, for HaDV2-infected insects was around 40% higher than that of their non-infected counterparts, which probably sets an upper limit to the potential impact of HaDV2 infection. Following Himler (2011), a simple population growth model based on these R_0_ estimates suggests that HaDV2 would spread to close to fixation in Bt-cotton growing areas in 20-30 generations (*Figure 3 —figure supplement 1*) (page 12, lines 295-298).

8) Ideal demonstration of causality: In the laboratory, this would involve infecting a virus-free strain of bollworm, and then curing the strain of the virus so that pre-infection, infection, and post-infection performance could be compared. Is this even possible in the laboratory? In the field?

Right, we also tried to compare pre-infection (HaDV-), infection (HaDV2 infecting) and post-infection (HaDV2+) with samples collected at 24h, 48h and 72h after HaDV2 exposure. We conducted an RNA sequencing experiment may provide some answers to this question. The principal component analysis of the transcriptome with differentially expressed gene data clearly distinguished HaDV2-positive from -negative individuals at three different time points: 24 h, 48 h and less so at 72 h after HaDV2 inoculation, suggesting that the HaDV2 has a major effect on the gene expression profiles of their hosts. However, we thought our results only stand for pre-infection and post-infection. It’s difficult to determine the samples which were being infected by HaDV2. See pages 9-10, lines 212-234 for details.

9) Evidence was presented in earlier work that densovirus infection can reduce baculovirus infection. Does a latent or active baculovirus infection have any prospects to reverse the effect of densovirus on tolerance of Bt-cotton?

Many thanks for the suggestion. The covert infection of NPV was naturally occurred (Kemp et al., 2011; Murillo et al., 2011; Williams et al. 2017). It’s interested to run a novel investigation to determine whether latent or active baculovirus infection has any prospects to reverse the effect of HaDV2 on tolerance of Bt by series bioassays.

Kemp EM, Woodward DT, Cory JS 2011. Detection of single and mixed covert baculovirus infections in eastern spruce budworm, Choristoneura fumiferana populations. J Invertebr Pathol 107: 202-205.

Murillo R, Hussey MS, Possee RD 2011. Evidence for covert baculovirus infections in a Spodoptera exigua laboratory culture. J Gen Virol 92: 1061-1070.

Williams T, Virto C, Murillo R, Caballero P 2017. Covert Infection of Insects by Baculoviruses. Front Microbiol 8: 1337.